# Temporal Difference Learning for Diffusion Models

**Qizhen Ying** [1]   **Yangchen Pan** [1]   **Victor Adrian Prisacariu** [1]   **Junfeng Wen** [2]

## Abstract

Diffusion models are typically trained with objectives that focus on local denoising targets at individual time steps (or adjacent pairs), which do not enforce consistency between predictions along the denoising trajectory. This lack of cross-time consistency can degrade performance, especially for few-step samplers. We introduce a temporal difference (TD) objective that penalizes inconsistency of the model's *multi-step* progress along the denoising path. By reformulating the diffusion process as a Markov reward process and casting denoising as a policy evaluation problem in reinforcement learning, we derive a unified TD approach that applies to both discrete- and continuous-time diffusion formulations. We further propose a principled sample-based reweighting method that stabilizes training. Empirically, we show that using our TD training can significantly improve sample quality measured by FID, with stronger advantages when the number of sampling steps is small, highlighting its practical utility under low-computation-budget scenarios. We provide ablation studies to justify our design choices, including pairwise loss reweighting, regularization weight, and one-step stride. Overall, our TD approach can be a general drop-in that enforces cross-time consistency and improves generation quality across different diffusion generative models.

## 1. Introduction

Diffusion models have become a standard tool for high-fidelity generative modeling across images, audio, and beyond (Ho et al., 2020; Song et al., 2021b; Karras et al.,

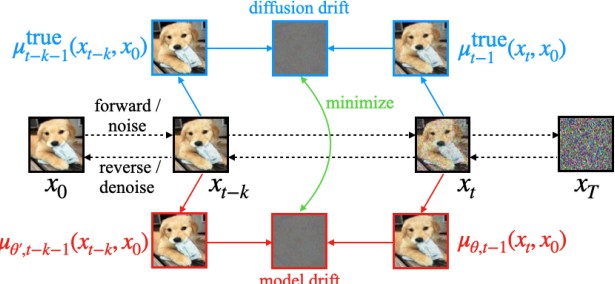

*Figure 1.* Overview of our algorithm. By matching the model drift (calculated from the posterior means) with diffusion drift, we enforce consistency over denoising time index/noise level.

2022). Despite impressive progress in *sampler* design (e.g., probability-flow ODE/DDIM (Song et al., 2021a;b), high-order solvers such as DPM-Solver (Lu et al., 2022) and UniPC (Zhao et al., 2023)) and *training-time accelerations* (e.g., progressive distillation (Salimans & Ho, 2022) and consistency-style learning (Song et al., 2023)), the dominant training paradigm still optimizes *single-time* reconstruction/noise-prediction losses.

Such single-time objectives do not explicitly require that predictions made at different noise levels form a *time-consistent* trajectory under the known forward corruption process. The resulting cross-time mismatch can accumulate along the denoising path and becomes particularly detrimental when the sampler uses few steps, i.e., a small number of function evaluations (NFE), where local errors have limited opportunity to average out (Song et al., 2021b; Karras et al., 2022). This motivates treating diffusion training through the lens of sequential decision making, where predictions at different timesteps must be consistent over multiple steps rather than only locally accurate.

Another line of recent work targets the same few-step sampling bottleneck by modifying the learned transport or denoising map itself. Shortcut models condition the network on both the current noise level and a desired step size, enabling the same model to take short or long denoising jumps at inference time (Frans et al., 2025). MeanFlow instead learns an average velocity field for one-step generative modeling, providing a flow-based alternative to modeling only instantaneous velocities (Geng et al., 2026). These approaches are closely related in motivation to fast sampling, but they primarily redesign the generator or its parameterization for

[1]Department of Engineering Science, University of Oxford, Oxford, United Kingdom [2]School of Computer Science, Carleton University, Ottawa, Canada. Correspondence to: Qizhen Ying <qizhen.ying@eng.ox.ac.uk>.

*Proceedings of the 43$^{rd}$ International Conference on Machine Learning*, Seoul, South Korea. PMLR 306, 2026. Copyright 2026 by the author(s).

one-/few-step generation. Our goal is complementary: we keep the base diffusion or consistency-training objective and add a TD regularizer that enforces cross-time consistency of posterior-mean drifts along the denoising trajectory.

Recent years have seen a surge of works that formulate diffusion sampling as a multi-step decision or control problem and apply reinforcement learning (RL) or control-based methods to optimize non-differentiable rewards. DDPO casts denoising as a Markov decision process (MDP) and shows policy-gradient updates can align text-to-image models with black-box objectives such as aesthetics and compressibility (Black et al., 2023). DPOK performs online RL with KL regularization to fine-tune diffusion models from human-trained reward functions, improving both alignment and fidelity (Fan et al., 2023). Adjoint Matching formulates reward fine-tuning for flow and diffusion models through stochastic optimal control, yielding a regression-style objective for reward-guided model improvement (Domingo i Enrich et al., 2025).

Other algorithmic variants include LOOP, which analyzes the efficiency–performance trade-off between REINFORCE and PPO and proposes a leave-one-out PPO scheme for diffusion fine-tuning (Gupta et al., 2025); and SEPO, which develops a policy-gradient method for discrete diffusion with theoretical justification and strong results across discrete generative tasks (Zekri & Boullé, 2025). Beyond images, RL fine-tuning has been applied to text-to-speech diffusion models via a loss-guided policy optimization objective (DLPO) (Chen et al., 2024). Other directions include self-play (SPIN-Diffusion), where a model competes with its past checkpoints to iteratively improve under a reward signal (Yuan et al., 2024), and forward-process RL that integrates reinforcement signals into flow-/score-matching objectives for online fine-tuning (Zheng et al., 2025).

Finally, at the theoretical level, Temporal Difference Flows connect TD learning with flow-based training, providing an RL interpretation of generative flows (Farebrother et al., 2025). Earlier work on generative TD learning proposed the $\gamma$-model framework, which reinterprets TD updates as a generative modeling problem for infinite-horizon prediction (Janner et al., 2020). While these methods highlight the synergy between RL and generative models, they focus either on flow-based models or on predictive state distributions.

A complementary strand leverages diffusion *as the policy class* for decision making in RL: Diffuser denoises entire trajectories to plan behaviors (Janner et al., 2022), Diffusion-QL represents policies with conditional diffusion models for offline RL (Wang et al., 2023), and hierarchical methods introduce subgoal-conditioned diffusion for long-horizon tasks (Li et al., 2023). These approaches focus on maximizing environmental returns in external tasks rather than developing new training regimes for diffusion models.

In contrast, our work performs *policy evaluation* over the denoising process itself. We reformulate diffusion as a Markov reward process (MRP) and introduce a TD objective that enforces cross-time consistency of predictions along the denoising path, unified across discrete- and continuous-time formulations. Unlike prior consistency models (CM) (Song et al., 2023) that require the reconstructions to agree over time, our method encourages the change in posterior means between two times to match the true diffusion drift, as illustrated in Figure 1. Furthermore, to stabilize optimization across different time pairs, we propose a principled sample-based loss reweighting scheme that equalizes loss scales. Rather than learning a new one-step generator or steering outputs through task-specific rewards, our TD formulation serves as a general-purpose training objective that improves fixed-NFE generation by aligning the model's internal temporal dynamics.

The rest of the paper is organized as follows. In Sec. 2, we review background and notation, including a unified two-time-mean form. Sec. 3 presents our TD objective (discrete derivation, unified form, design rules, etc.). We present the experiment results in Sec. 4 and discuss our method in Sec. 5. The appendices include implementation details, scheduler definitions, and additional proofs.

## 2. Background and Notation

Diffusion models corrupt data with a forward (noising) process and learn a denoiser or score function that inverts it (Ho et al., 2020; Song et al., 2021b). Discrete-time formulations such as DDPM (Ho et al., 2020; Nichol & Dhariwal, 2021) and the deterministic DDIM (Song et al., 2021a) provide simple training objectives and flexible sampling schedules, while continuous-time formulations based on differential equations (ODE/SDE) unify these views and support principled SDE samplers (Song et al., 2021b). To address the complex design space of diffusion models, EDM (Karras et al., 2022) modularizes the framework, refines several design choices (e.g., noise grids, preconditioning, loss weighting, etc.) and achieves strong empirical results with few-step sampling. In this work, we adopt these foundations and focus on enforcing *cross-time consistency* at training time.

### 2.1. Unified Two-Time Posterior Mean

Let $t$ denote the time index: a discrete time $t \in \{0, \ldots, T\}$ (DDPM/DDIM), or a continuous time $t \in [0, T]$ (ODE/SDE). Let $\tau < t$ be an *earlier (cleaner)* time step. Across families, the mean of the posterior $q(\boldsymbol{x}_\tau | \boldsymbol{x}_t, \boldsymbol{x}_0)$ admits the *same linear form* (derived in Appendix B)

$$\boldsymbol{\mu}_\tau^{\text{true}}(\boldsymbol{x}_t, \boldsymbol{x}_0) = A_{t,\tau}\, \boldsymbol{x}_0 + \kappa_{t,\tau}\, \boldsymbol{x}_t, \tag{1}$$

where $\boldsymbol{x}_0$ is a clean datum (e.g., original image) and $\boldsymbol{x}_t$ is a noisy sample at level $t$. Family-specific choices for

| Process | $A_{t,\tau}$ | $\kappa_{t,\tau}$ |
|---|---|---|
| DDPM | $\dfrac{\sqrt{\bar{\alpha}_\tau}\,(1-\alpha_t)}{1-\bar{\alpha}_t}$ | $\dfrac{\sqrt{\alpha_t}\,(1-\bar{\alpha}_\tau)}{1-\bar{\alpha}_t}$ |
| DDIM | $\sqrt{\bar{\alpha}_\tau} - \sqrt{\dfrac{1-\bar{\alpha}_\tau}{1-\bar{\alpha}_t}}\,\sqrt{\bar{\alpha}_t}$ | $\sqrt{\dfrac{1-\bar{\alpha}_\tau}{1-\bar{\alpha}_t}}$ |
| VP-SDE | $\alpha(\tau) - \kappa_{t,\tau}\,\alpha(t)$ | $\dfrac{\alpha(t)}{\alpha(\tau)}\cdot\dfrac{1-\alpha(\tau)^2}{1-\alpha(t)^2}$ |
| VE-SDE / EDM | $1 - \dfrac{\sigma(\tau)^2}{\sigma(t)^2}$ | $\dfrac{\sigma(\tau)^2}{\sigma(t)^2}$ |
| CM | $1 - \dfrac{\sigma(\tau)}{\sigma(t)}$ | $\dfrac{\sigma(\tau)}{\sigma(t)}$ |

*Table 1.* Unified two-time posterior mean coefficients in $\boldsymbol{\mu}_\tau^{\text{true}}(\boldsymbol{x}_t, \boldsymbol{x}_0) = A_{t,\tau}\boldsymbol{x}_0 + \kappa_{t,\tau}\boldsymbol{x}_t$ for DDPM (Ho et al., 2020), DDIM (Song et al., 2021a), VP/VE-SDE (Song et al., 2021b), EDM (Karras et al., 2022), and CM (Song et al., 2023). For DDPM/DDIM, we often use the adjacent-step $\tau = t-1$ (though the formula holds for any $\tau < t$). For VP/VE/EDM/CM, $(t, \tau)$ are continuous time indices (equivalently, noise levels) selected by the sampler. See Appendix B for definitions and derivations.

$(A_{t,\tau}, \kappa_{t,\tau})$ are summarized in Table 1. This linear form will allow us to define a surrogate mean by predicting $\boldsymbol{x}_0$ using a model $\boldsymbol{x}_{\theta,0}(\boldsymbol{x}_t; t) \approx \boldsymbol{x}_0$ parametrized by $\theta$ (e.g., the $D_\theta$ and $f_\theta$ in the following paragraphs). This will be the main strategy for our TD objective (Sec.3).

**EDM-style preconditioning.** EDM (Karras et al., 2022) wraps a raw network $F_\theta$ in the preconditioned denoiser as[1]

$$D_\theta(\boldsymbol{x}; t) = c_{\text{skip}}^{\text{EDM}}(t)\,\boldsymbol{x} + c_{\text{out}}^{\text{EDM}}(t)\,F_\theta\big(c_{\text{in}}(t)\,\boldsymbol{x}; c_{\text{noise}}(t)\big), \tag{2}$$

and trains this model with a weighted regression

$$\mathcal{L}_{\text{EDM}} = \mathbb{E}_{t,\boldsymbol{x}_0,\boldsymbol{x}_t}\Big[w(t)\,\big\|D_\theta(\boldsymbol{x}_t; t) - \boldsymbol{x}_0\big\|_2^2\Big]. \tag{3}$$

The specific choices of $\{c_{\text{skip}}^{\text{EDM}}, c_{\text{out}}^{\text{EDM}}, c_{\text{in}}, c_{\text{noise}}, w(t)\}$ equalize the effective variance across noise levels and improve optimization conditioning (Karras et al., 2022). We will develop a weighting to achieve a similar effect later, and combine Equation (3) with our TD objective.

**Preconditioning for consistency models.** Similarly to EDM, CM (Song et al., 2023) parametrizes a prediction model $f_\theta(\boldsymbol{x}_t; t)$ to predict $\boldsymbol{x}_\epsilon$ using any $t \in [\epsilon, T]$ where $\epsilon > 0$ is a small number. To ensure that the boundary condition $f_\theta(\boldsymbol{x}_\epsilon; \epsilon) = \boldsymbol{x}_\epsilon$ is satisfied by construction, CM adopts a preconditioned form

$$f_\theta(\boldsymbol{x}; t) = c_{\text{skip}}^{\text{CM}}(t)\,\boldsymbol{x} + c_{\text{out}}^{\text{CM}}(t)\,F_\theta(\boldsymbol{x}; t), \tag{4}$$

satisfying $c_{\text{skip}}^{\text{CM}}(\epsilon) = 1$, $c_{\text{out}}^{\text{CM}}(\epsilon) = 0$, which is a boundary-shifted variant of EDM's parameterization. When training from scratch (i.e., consistency training or CT), it optimizes

$$\mathcal{L}_{\text{CT}} = \mathbb{E}\Big[w(t)\,\big\|f_\theta(\boldsymbol{x}_{t+1}; t+1) - f_{\theta'}(\boldsymbol{x}_t, t)\big\|_2^2\Big], \tag{5}$$

---

[1] Note that EDM adopts a noise scheme instead of a time scheme, but we use time here as they are equivalent and it can facilitate later discussion.

where $\theta'$ is updated via exponential moving average (EMA).

## 2.2. Temporal Difference Learning

In this work, we will recast training diffusion models as policy evaluation problems in reinforcement learning (RL) (Pan et al., 2025). A finite-horizon Markov Reward Process (MRP) (Szepesvári, 2022) is characterized by a tuple $(\mathcal{X}, r_t, P_t, T)$ where $\mathcal{X}$ is the (common) state space, $r_t : \mathcal{X} \times \mathcal{X} \mapsto \mathbb{R}$ is the reward function at time $t$, $P_t : \mathcal{X} \mapsto \Delta(\mathcal{X})$ is the transition kernel at time $t$ and $T$ is the length of the episode. However, to match the time notation in diffusion models, here we let the MRP start from step $t = T$ and traverse *backward* to $t = 0$. Specifically, when transitioning from $\boldsymbol{x}_t$ to $\boldsymbol{x}_{t-1}$ according to $P_t(\cdot|\boldsymbol{x}_t)$, we receive a reward of $r_{t-1}(\boldsymbol{x}_t, \boldsymbol{x}_{t-1})$. Let $g_t := \sum_{i=1}^{t} r_{t-i}$ denote the MRP return collected from time $t$ until the terminal time 0, where $r_{t-i}$ is short for $r_{t-i}(x_{t-i+1}, x_{t-i})$. The state value function is then defined as the expected return when starting from a given state:

$$v_t(\boldsymbol{x}) := \mathbb{E}[g_t \mid \boldsymbol{x}_t = \boldsymbol{x}] := \mathbb{E}\left[\sum_{i=1}^{t} r_{t-i} \,\middle|\, \boldsymbol{x}_t = \boldsymbol{x}\right] \tag{6}$$

where the expectation is taken over $P_i$ for $0 < i \leq t$.

One can use temporal difference (TD) learning (Sutton & Barto, 2018) to find $v_t$. Given some parametrized functions $v_{\theta,t}$, the TD error of the transition $(\boldsymbol{x}_t, \boldsymbol{x}_{t-1})$ is

$$\delta_t := r_{t-1} + v_{\theta,t-1}(\boldsymbol{x}_{t-1}) - v_{\theta,t}(\boldsymbol{x}_t). \tag{7}$$

By minimizing the TD error using samples from the MRP and semi-gradient descent, $v_{\theta,t}$ will converge to the true $v_t$ (Bertsekas & Tsitsiklis, 1996; Tsitsiklis & Van Roy, 1996).

## 3. Temporal Difference Learning for Diffusion Models

This section develops TD training for diffusion models. We first derive the TD loss on a discrete time index grid, then lift it to a unified formulation that applies to both discrete-(DDPM/DDIM) and continuous-time (VP/VE/EDM) families. Whenever a posterior mean is needed, we *do not* restate family-specific formulas – rather, we directly use the unified form in Equation (1).

### 3.1. Discrete Time: MRP Formulation for DDPM

**MRP specifications on the time index grid.** Following Sec. 2.2, we define an MRP, traversing backward in time from $t = T$ to $t = 0$. The construction is tied to a clean sample $\boldsymbol{x}_0$: the rewards and posterior transitions below are defined conditionally on a clean $\boldsymbol{x}_0$ sampled from the data distribution. In other words, $\boldsymbol{x}_0$ acts as an episode-level

context throughout the derivation, and we omit it when no confusion arises. For training objectives, $x_0 \sim q_{\text{data}}$ is a random variable, as shown later in Equation (16); within a sampled episode, the realized $x_0$ is treated as fixed. In our context of diffusion model training, rewards, returns, and values are all data-space vector-valued quantities, which corresponds to a multiple-reward setting. The MRP is specified as follows:

- *State space:* $\mathcal{X}$ is the data space. In the case of image generation, $\mathcal{X}$ is the set of images.
- *Reward function:* $r_{t-1} := r_{t-1}(x_t, x_{t-1}) := \mu_{t-1}^{\text{true}}(x_t, x_0) - \mu_{t-2}^{\text{true}}(x_{t-1}, x_0)$ is the posterior mean difference. Note that $r_{t-1}$ is a vector in the data space, as opposed to a scalar reward common in reinforcement learning. In the final step when transitioning from $x_1$ to $x_0$, the reward is defined as $r_0 := \mu_0^{\text{true}}(x_1, x_0) - x_0 = \mathbf{0}$. The last equation is because $\mu_0^{\text{true}}(x_1, x_0)$, the conditional mean, must be $x_0$ *given* $x_0$.
- *Transition kernel:* $P_t(x_{t-1} \mid x_t) := q(x_{t-1}|x_t, x_0)$ is induced by the posterior of the predecessor in the diffusion process (stochastic in $x_{t-1}$ through the forward coupling).

**Return, value, TD(0) and $k$-step return.** In this diffusion MRP setup, the cumulative reward has a simple form. Based on the definition of reward, the return $g_t$ (conditioned on $x_0$) then equals the displacement of the posterior mean from the data:

$$g_t \mid x_0 = \mu_{t-1}^{\text{true}}(x_t, x_0) - x_0, \qquad (8)$$

It satisfies the usual return recursion, here written backward in diffusion time, $g_t = r_{t-1} + g_{t-1}$. Moreover, it ensures that $g_1 = \mu_0^{\text{true}}(x_1, x_0) - x_0 = x_0 - x_0 = \mathbf{0}$. Because the posterior mean in Equation (8) is determined once $(x_t, x_0)$ is known, the corresponding value function reduces to this conditional return:

$$v_t(x_t) := \mathbb{E}[g_t \mid x_t, x_0] = \mu_{t-1}^{\text{true}}(x_t, x_0) - x_0. \quad (9)$$

Here, the subscript $t$ in $v_t$ denotes the diffusion-time index of the quantity being estimated (namely $g_t$), while $x_t$ is the conditioning state in the expectation. We approximate $v_t$ using the preconditioned denoiser, e.g., (2) or (4) earlier, as the model $x_{\theta,0}$:

$$v_t(x_t) \approx v_{\theta,t}(x_t) := \mu_{\theta,t-1}(x_t) - x_0 \qquad (10)$$

$$\mu_{\theta,t-1}(x_t) := A_{t,t-1} \, x_{\theta,0}(x_t; t) + \kappa_{t,t-1} \, x_t. \qquad (11)$$

We learn $\theta$ by constructing a bootstrap target with fixed parameters. The "next"-state's value is estimated by

$$v_{t-1}(x_{t-1}) \approx v_{\theta',t-1}(x_{t-1}) := \mu_{\theta',t-2}(x_{t-1}) - x_0 \quad (12)$$

$$\mu_{\theta',t-2}(x_{t-1}) = A_{t-1,t-2} \, x_{\theta',0}(x_{t-1}; t-1)$$
$$+ \kappa_{t-1,t-2} \, x_{t-1}, \qquad (13)$$

where $\theta'$ is the fixed (stop-gradient) target network's parameters, updated using EMA. By the definition of the reward, the bootstrap target is

$$r_{t-1} + v_{\theta',t-1}(x_{t-1}) = \mu_{t-1}^{\text{true}}(x_t, x_0) - \mu_{t-2}^{\text{true}}(x_{t-1}, x_0)$$
$$+ \mu_{\theta',t-2}(x_{t-1}) - x_0. \qquad (14)$$

Combining it with Equation (10) gives the TD error

$$\delta_t := r_{t-1} + v_{\theta',t-1}(x_{t-1}) - v_{\theta,t}(x_t)$$
$$= \underbrace{[\mu_{t-1}^{\text{true}}(x_t, x_0) - \mu_{t-2}^{\text{true}}(x_{t-1}, x_0)]}_{\text{one-step diffusion drift}}$$
$$- \underbrace{[\mu_{\theta,t-1}(x_t) - \mu_{\theta',t-2}(x_{t-1})]}_{\text{one-step model drift}}. \qquad (15)$$

The TD(0) objective is then

$$\mathcal{L}_{\text{TD}(0)} := \mathbb{E}_{t,x_0,x_{t-1},x_t} \left[ \|\delta_t\|_2^2 \right], \qquad (16)$$

where $t \sim \mathcal{U}\{2, \ldots, T\}$, $x_0 \sim q_{\text{data}}$, $x_{t-1} \sim q(x_{t-1} \mid x_0)$, and $x_t \sim q(x_t \mid x_{t-1})$.

Minimizing the TD error can be interpreted as aligning diffusion progression across time steps as shown in Equation (15). If the true mean has drifted in one step, the model should shift in the same way, thus enforcing *consistency* between time steps. The same derivation applies to DDIM (Song et al., 2021a) by substituting its $(A_{t,t-1}, \kappa_{t,t-1})$ from Table 1.

We can also use $k$-step return as the bootstrap target. Keep expanding Equation (14) over $k$ steps gives the following $k$-step objective

$$\mathcal{L}_{\text{TD}}^{(k)} := \mathbb{E}\left[ \| \underbrace{[\mu_{t-1}^{\text{true}}(x_t, x_0) - \mu_{t-k-1}^{\text{true}}(x_{t-k}, x_0)]}_{k\text{-step diffusion drift}} \right.$$
$$\left. - \underbrace{[\mu_{\theta,t-1}(x_t) - \mu_{\theta',t-k-1}(x_{t-k})]}_{k\text{-step model drift}} \|_2^2 \right]. \quad (17)$$

where the expectation is over $t \sim \mathcal{U}\{k+1, T\}$, $x_0 \sim q_{\text{data}}(x_0)$, $x_{t-k} \sim q(x_{t-k}|x_0)$, $x_t \sim q(x_t|x_{t-k})$. The matching process is illustrated in Figure 1.

**Aligning model with posterior.** Note that there is no need to bootstrap when a $t \le k$ is sampled because we do know the rest of the episode and the corresponding true return. In this case, the TD loss actually reduces to posterior mean matching (cf. (8) and (10)), similar to the DDPM loss

$$\mathcal{L}_{\text{DDPM}} = \mathbb{E}\left[ w(t) \|\mu_{t-1}^{\text{true}}(x_t, x_0) - \mu_{\theta,t-1}(x_t)\|_2^2 \right], \quad (18)$$

with a specific weighting $w(t)$ defined by DDPM. In practice, we combine the TD loss with the DDPM loss even

when $t > k$ to facilitate training and speed up convergence, and the final objective is

$$\mathcal{L}_{\text{TD+DDPM}}^{(k)} = \mathcal{L}_{\text{TD}}^{(k)} + \lambda \mathcal{L}_{\text{DDPM}}, \quad (19)$$

where $\lambda > 0$ is a hyper-parameter. In other words, if a time $t > k$ is sampled during training, the TD loss can be used and we optimize $\mathcal{L}_{\text{TD+DDPM}}^{(k)}$. Otherwise, we optimize $(1 + \lambda)\mathcal{L}_{\text{DDPM}}$ since the TD loss reduces to the DDPM loss in this case and the $(1 + \lambda)$ factor maintains the scale of the objective.

### 3.2. Discrete and Continuous Time: A Unified TD Objective

The derivations based on discrete time steps above can be easily extended to continuous-time scenarios with ODE/SDE thanks to the unified mean (Equation (1) from Sec. 2.1).

Instead of matching drifts that are $k$-step away in the discrete case, here we pick two time indices $t, t' \in [0, T]$. Time $t$ (resp. $t'$) induces a true posterior mean for an earlier time $\tau < t$ (resp. $\tau' < t'$). Their corresponding means can be expressed as

$$\boldsymbol{\mu}_\tau^{\text{true}}(\boldsymbol{x}_t, \boldsymbol{x}_0) = A_{t,\tau} \boldsymbol{x}_0 + \kappa_{t,\tau} \boldsymbol{x}_t \quad (20)$$

$$\boldsymbol{\mu}_{\tau'}^{\text{true}}(\boldsymbol{x}_{t'}, \boldsymbol{x}_0) = A_{t',\tau'} \boldsymbol{x}_0 + \kappa_{t',\tau'} \boldsymbol{x}_{t'}. \quad (21)$$

For the discrete-time case (Equation (17)), $\boldsymbol{\mu}_{t-1}^{\text{true}}(\boldsymbol{x}_t, \boldsymbol{x}_0)$ is the posterior mean in the previous time step (hence the subscript $t - 1$). In analogy, $\tau$ here can be considered as the "previous time step" of $t$ in the continuous case. In our experiments, we set $\tau' < t' < \tau < t$ with *span* $k := t - t'$ and *stride* $\Delta := t' - \tau' = t - \tau < k$, imitating the discrete case. Accordingly, the model is defined as

$$\boldsymbol{\mu}_{\theta,\tau}(\boldsymbol{x}_t) := A_{t,\tau} \boldsymbol{x}_{\theta,0}(\boldsymbol{x}_t; t) + \kappa_{t,\tau} \boldsymbol{x}_t, \quad (22)$$

where $\boldsymbol{x}_{\theta,0}$ predicting $\boldsymbol{x}_0$, instantiated as $D_\theta$ for EDM (2) or (4) for CM parameterizations (Sec. 2.1). The TD loss for continuous time reads

$$\mathcal{L}_{\text{TD}}^{\text{cont}} = \mathbb{E}_{\boldsymbol{x}_0, t, t', \boldsymbol{x}_t, \boldsymbol{x}_{t'}} \Big[ \big\| \big[ \boldsymbol{\mu}_\tau^{\text{true}}(\boldsymbol{x}_t, \boldsymbol{x}_0) - \boldsymbol{\mu}_{\tau'}^{\text{true}}(\boldsymbol{x}_{t'}, \boldsymbol{x}_0) \big]$$
$$- \big[ \boldsymbol{\mu}_{\theta,\tau}(\boldsymbol{x}_t) - \boldsymbol{\mu}_{\theta',\tau'}(\boldsymbol{x}_{t'}) \big] \big\|_2^2 \Big]. \quad (23)$$

Inspired by prior work, it is preferable to include a weighting scheme $w_{\text{TD}}(t, t')$ for loss per sample within the expectation of Equation (23) to avoid drastic changes in gradient magnitudes across time steps. The weighting scheme then depends on the parametrization of the prediction model and the training strategies of the base algorithm. In the following, we will incorporate our TD objective into two widely used continuous-time training paradigms, EDM and CT, and show specific recipes for stable training.

### 3.3. Training Recipes: TD+EDM and TD+CT

**Weighting for EDM parameterization.** Here we expand the TD error (23) by substituting (22) into it

$$\boldsymbol{\delta}_{t,t'} := \big[ \boldsymbol{\mu}_\tau^{\text{true}}(\boldsymbol{x}_t, \boldsymbol{x}_0) - \boldsymbol{\mu}_{\tau'}^{\text{true}}(\boldsymbol{x}_{t'}, \boldsymbol{x}_0) \big]$$
$$- \big[ \boldsymbol{\mu}_{\theta,\tau}(\boldsymbol{x}_t) - \boldsymbol{\mu}_{\theta',\tau'}(\boldsymbol{x}_{t'}) \big] \quad (24)$$
$$= \big[ A_{t,\tau} \boldsymbol{x}_0 - A_{t',\tau'} \boldsymbol{x}_0 \big]$$
$$- \big[ A_{t,\tau} \boldsymbol{x}_{\theta,0}(\boldsymbol{x}_t, t) - A_{t',\tau'} \boldsymbol{x}_{\theta',0}(\boldsymbol{x}_{t'}, t') \big]. \quad (25)$$

When using the EDM preconditioned denoiser $D_\theta$ (2) as $\boldsymbol{x}_{\theta,0}$, the TD error can be decomposed further by using the raw network $F_\theta$

$$\boldsymbol{\delta}_{t,t'} = \big[ A_{t,\tau} \boldsymbol{x}_0 - A_{t',\tau'} \boldsymbol{x}_0 \big]$$
$$- \big[ A_{t,\tau} c_{\text{skip}}^{\text{EDM}}(t) \boldsymbol{x}_t - A_{t',\tau'} c_{\text{skip}}^{\text{EDM}}(t') \boldsymbol{x}_{t'} \big] \quad (26)$$
$$- \big[ A_{t,\tau} c_{\text{out}}^{\text{EDM}}(t) F_{\theta,t}(\boldsymbol{x}_t) - A_{t',\tau'} c_{\text{out}}^{\text{EDM}}(t') F_{\theta',t'}(\boldsymbol{x}_{t'}) \big].$$

where we use $F_{\theta,t}(\boldsymbol{x}_t) = F_\theta(c_{\text{in}}(t)\boldsymbol{x}_t; c_{\text{noise}}(t))$ for short. Following EDM, which balances the per-sample losses for the raw model $F_\theta$, here we also equalize the loss scale for $F_\theta$ and $F_{\theta'}$. Minimizing the TD error effectively minimizes the normalized errors $\boldsymbol{e}_t^{\text{EDM}}, \boldsymbol{e}_{t'}^{\text{EDM}}$

$$\boldsymbol{e}_t^{\text{EDM}} := \frac{\boldsymbol{x}_0 - c_{\text{skip}}^{\text{EDM}}(t) \boldsymbol{x}_t}{c_{\text{out}}^{\text{EDM}}(t)} - F_{\theta,t}(\boldsymbol{x}_t),$$
$$\boldsymbol{e}_{t'}^{\text{EDM}} := \frac{\boldsymbol{x}_0 - c_{\text{skip}}^{\text{EDM}}(t') \boldsymbol{x}_{t'}}{c_{\text{out}}^{\text{EDM}}(t')} - F_{\theta',t'}(\boldsymbol{x}_{t'}). \quad (27)$$

rescaled by $A_{t,\tau} c_{\text{out}}^{\text{EDM}}(t)$ (resp. $A_{t',\tau'} c_{\text{out}}^{\text{EDM}}(t')$). That is, $\boldsymbol{\delta}_{t,t'} = \mathcal{B} \, \boldsymbol{e}_{t,t'}^{\text{EDM}}$ where

$$\mathcal{B} := \big[ A_{t,\tau} c_{\text{out}}^{\text{EDM}}(t) I, \ -A_{t',\tau'} c_{\text{out}}^{\text{EDM}}(t') I \big] \in \mathbb{R}^{d \times 2d},$$
$$\boldsymbol{e}_{t,t'}^{\text{EDM}} := \begin{bmatrix} \boldsymbol{e}_t^{\text{EDM}} \\ \boldsymbol{e}_{t'}^{\text{EDM}} \end{bmatrix} \in \mathbb{R}^{2d}. \quad (28)$$

Then we can see that

$$\|\boldsymbol{\delta}_{t,t'}\|_2^2 = \|\mathcal{B} \, \boldsymbol{e}_{t,t'}\|_2^2 \leq \|\mathcal{B}\|_2^2 \|\boldsymbol{e}_{t,t'}^{\text{EDM}}\|_2^2 \quad (29)$$
$$= \Big( A_{t,\tau}^2 c_{\text{out}}^{\text{EDM}}(t)^2 + A_{t',\tau'}^2 c_{\text{out}}^{\text{EDM}}(t')^2 \Big) \|\boldsymbol{e}_{t,t'}^{\text{EDM}}\|_2^2.$$

Since $\|\boldsymbol{e}_{t,t'}^{\text{EDM}}\|_2^2$ is the normalized error w.r.t. the raw models $F_\theta, F_{\theta'}$, it would be helpful to set a uniform scale that is not affected by the choice of time indices, which leads to the following pairwise weighting:

$$w_{\text{TD}}^{\text{EDM}}(t, t') = \frac{1}{A_{t,\tau}^2 c_{\text{out}}^{\text{EDM}}(t)^2 + A_{t',\tau'}^2 c_{\text{out}}^{\text{EDM}}(t')^2}. \quad (30)$$

Then $w_{\text{TD}}(t, t')\|\boldsymbol{\delta}_{t,t'}\|_2^2 \leq \|\boldsymbol{e}_{t,t'}^{\text{EDM}}\|_2^2$, and the weighted TD objective is

$$\mathcal{L}_{\text{wTD+EDM}}^{\text{cont}} = \mathbb{E}_{\boldsymbol{x}_0, t, t'} \Big[ w_{\text{TD}}^{\text{EDM}}(t, t') \|\boldsymbol{\delta}_{t,t'}\|_2^2 \Big]. \quad (31)$$

**Weighting for CM parameterization.** Similarly, we can apply CM's model $f_\theta$ (4) to Equation (25). Instead of balancing losses for the raw model $F_\theta$, consistency training (CT) applies a uniform weighting $w(t) \equiv 1$ in (5) (Song et al., 2023), equalizing the losses for $f_\theta$ directly. Following this practice, we focus on the errors for $f_\theta$

$$\boldsymbol{e}_t^{\mathrm{CT}} := \boldsymbol{x}_0 - f_\theta(\boldsymbol{x}_t; t), \quad \boldsymbol{e}_{t'}^{\mathrm{CT}} := \boldsymbol{x}_0 - f_{\theta'}(\boldsymbol{x}_{t'}; t'). \quad (32)$$

Substituting these into Equation (25) and following the same derivation, we bound the TD error as

$$\|\boldsymbol{\delta}_{t,t'}\|_2^2 \leq \left(A_{t,\tau}^2 + A_{t',\tau'}^2\right) \|\boldsymbol{e}_{t,t'}^{\mathrm{CT}}\|_2^2. \quad (33)$$

Hence, we have the CT-specific pairwise weight:

$$w_{\mathrm{TD}}^{\mathrm{CT}}(t,t') := \frac{1}{A_{t,\tau}^2 + A_{t',\tau'}^2}. \quad (34)$$

The general weighted TD objective is then

$$\mathcal{L}_{\mathrm{wTD+CT}}^{\mathrm{cont}} = \mathbb{E}_{\boldsymbol{x}_0, t, t'}[w_{\mathrm{TD}}^{\mathrm{CT}}(t,t')\|\boldsymbol{\delta}_{t,t'}\|_2^2]. \quad (35)$$

**Aligning model with posterior.** During training, $t$ is sampled within $[0, T]$, which can make $\tau'$ invalid given fixed span $k$ and stride $\Delta$. Specifically, when $\tau' = t - k - \Delta$ falls outside the valid time window (or equivalently, when the corresponding noise level goes below the minimum noise of the sampler), the TD loss reduces to a mean matching objective, similar to the discrete-time case. Therefore, for a base continuous-time algorithm (e.g., EDM or CT), we can incorporate our TD loss into the original loss, and optimize an objective based on the sampled time-step or noise-level. For EDM, our objective is

$$\mathcal{L}_{\mathrm{TD+EDM}} = \mathcal{L}_{\mathrm{wTD+EDM}}^{\mathrm{cont}} + \lambda\mathcal{L}_{\mathrm{EDM}}, \quad (36)$$

when $\tau'$ is valid, and $(1 + \lambda)\mathcal{L}_{\mathrm{EDM}}$ otherwise. Similarly, for CT, our objective is

$$\mathcal{L}_{\mathrm{TD+CT}} = \mathcal{L}_{\mathrm{wTD+CT}}^{\mathrm{cont}} + \lambda\mathcal{L}_{\mathrm{CT}}. \quad (37)$$

when $\tau'$ is valid, and $(1 + \lambda)\mathcal{L}_{\mathrm{CT}}$ otherwise.

Our method is flexible in that, given a base continuous-time algorithm, we can incorporate our TD loss (23) into the base loss $\mathcal{L}_{\mathrm{base}}$ depending on the original algorithm and its model parametrization for predicting $\boldsymbol{x}_0$. The procedure is summarized in Algorithm 1.

**Noise-grid index mapping and TD pairing.** In implementation, we parameterize TD time indices using the sampler's noise grid of EDM (Karras et al., 2022):

$$\sigma(i) = \left(\sigma_{\max}^{1/\rho} + \frac{i}{N-1}\left(\sigma_{\min}^{1/\rho} - \sigma_{\max}^{1/\rho}\right)\right)^\rho, \quad (38)$$

where $i \in \{0, \ldots, N-1\}$, $N$ is the grid size, $\rho$ is the schedule exponent, and $(\sigma_{\min}, \sigma_{\max})$ are the upper and lower bounds of the sampler noise level. Given a noise level $\sigma$ within $[\sigma_{\min}, \sigma_{\max}]$, we identify its corresponding (possibly non-integer) index $i$ from Equation (38) and treat it as the TD time index $t$. We then set[2] $t' = t + k$ and define $\tau = t + \Delta$ and $\tau' = \tau + k$ to construct the TD pair in Equation (23). We apply TD only when $\sigma(\tau') \in [\sigma_{\min}, \sigma_{\max}]$; otherwise we fall back to the base loss mentioned above.

---

**Algorithm 1** TD training (general recipe)

---

**Input**: Preconditioned denoiser $\boldsymbol{x}_{\theta,0}$ (e.g., $D_\theta$ for EDM or $f_\theta$ for CT); target network $\theta'$; noise grid $\sigma(i)$; span $k$; stride $\Delta$; mixing coefficient $\lambda$; base loss $\mathcal{L}_{\mathrm{base}}$ (EDM or CT); pairwise TD weight $w_{\mathrm{TD}}(\cdot, \cdot)$.

1: Sample $\boldsymbol{x}_0 \sim q_{\mathrm{data}}(\boldsymbol{x}_0)$ and select a grid/noise index $t$ according to the base method.
2: Compute the base loss $\mathcal{L}_{\mathrm{base}}$ (EDM or CT) using $\boldsymbol{x}_{\theta,0}$
3: **if** $t \leq N - 1 - k - \Delta$ **then**
4:     Compute the TD error $\boldsymbol{\delta}_{t,t'}$.
5:     $\mathcal{L}_{\mathrm{wTD}} = w_{\mathrm{TD}}(t,t')\|\boldsymbol{\delta}_{t,t'}\|_2^2$  (EDM: (30); CT: (34))
6:     $\mathcal{L} = \mathcal{L}_{\mathrm{wTD}} + \lambda\mathcal{L}_{\mathrm{base}}$
7: **else**
8:     $\mathcal{L} = (1 + \lambda)\mathcal{L}_{\mathrm{base}}$
9: **end if**
10: Perform gradient descent on $\mathcal{L}$ and update $\theta'$ using exponential moving average.

---

## 4. Empirical Study

We evaluate our TD objective with (i) EDM training (TD+EDM) and (ii) Consistency model training (TD+CT).

**Experimental Setup.** In all experiments, we utilize the probability-flow ODE with the Heun integrator, where the NFE is *NFE = 2 × steps − 1*. We report the *last-15% FID-50k*, defined as the average FID-50k over the last 15% of evaluation checkpoints to ensure a stable performance measure. For TD+EDM, unless otherwise stated, we use the default TD setup selected by the EDM ablations: $\Delta = 0.25$, $k = 1$, $\lambda = 0.5$, and the TD pairwise weighting proposed in Section 3.3. See App. A for implementation details.

### 4.1. Performance and Applicability

**TD+EDM.** We first apply our TD training with EDM and evaluate performance across different inference steps under the standard EDM configuration (Karras et al., 2022) (App. A.1). Table 2 reports results on three benchmarks:

---

[2]We use $t' = t + k$ instead of $t' = t - k$ because then $t'$ will correspond to a smaller noise level, thus closer to the real data as common in the ODE/SDE formulations ($t = 0$). In noise grid larger $i$ corresponding to smaller noise level.

*Table 2.* Cross-dataset comparison under the same TD+EDM setting. FID-50k↓ (last-15% average).

| Dataset | Steps | TD+EDM | EDM |
|---|---|---|---|
| Cond. CIFAR-10 (32×32) | 9 | 3.799 | **3.703** |
| | 12 | **2.270** | 2.365 |
| | 15 | **2.235** | 2.311 |
| | 18 | **2.129** | 2.170 |
| AFHQv2 (64×64) | 9 | **5.935** | 5.985 |
| | 12 | 4.108 | **4.060** |
| | 15 | **3.554** | 3.588 |
| | 18 | **3.386** | 3.402 |
| FFHQ (64×64) | 9 | **7.463** | 7.829 |
| | 12 | **4.400** | 4.499 |
| | 15 | **3.564** | 3.695 |
| | 18 | **3.246** | 3.370 |

class-conditional CIFAR-10 (32×32), AFHQv2 (64×64), and FFHQ (64×64). Across these datasets, TD+EDM matches or improves upon the EDM baseline in the moderate few-step regime (12–18 steps). On FFHQ (and on AFHQv2 for most step budgets) specifically, our method allows the model to maintain higher fidelity than the baseline, which relies solely on local denoising targets. These results suggest that by aligning the model drift and diffusion drift, TD training makes the model more robust to the larger discretization intervals inherent in few-step solvers.

**TD+CT.** We further evaluate TD training within the consistency training (CT) framework, following the same CT baseline setup and evaluation setting in consistency models (Song et al., 2023) (App. A.2). For TD+CT, we also sweep the TD-specific hyperparameters, i.e., $\lambda$ and $\Delta$, over the same ranges as the EDM ablations. The best setting is $\lambda = 0.5$, $\Delta = 0.5$ (one-step stride = 1/2), with $k = 1$, which we use for both AFHQv2 (64×64) and FFHQ (64×64). As shown in Table 3, TD+CT improves one-step FID on both benchmarks. On AFHQv2, FID improves from 12.97 to 12.87, and on FFHQ, FID improves from 19.45 to 15.93.

The advantage of our TD objective is more evident by the learning curves and the multi-step sampling profiles. As illustrated in Figure 2, TD+CT consistently exhibits better convergence and lower FID across different sampling steps for FFHQ throughout the training process, while remaining competitive or better on AFHQ.

*Table 3.* TD+CT vs. CT baseline under one-step sampling (steps= 1, NFE= 1). FID-50k↓ (last-15% average over evaluation checkpoints).

| Dataset (setting) | Step | CT | TD+CT |
|---|---|---|---|
| AFHQv2 (64×64), adaptive $N$ | 1 | 12.97 | **12.87** |
| FFHQ (64×64), adaptive $N$ | 1 | 19.45 | **15.93** |

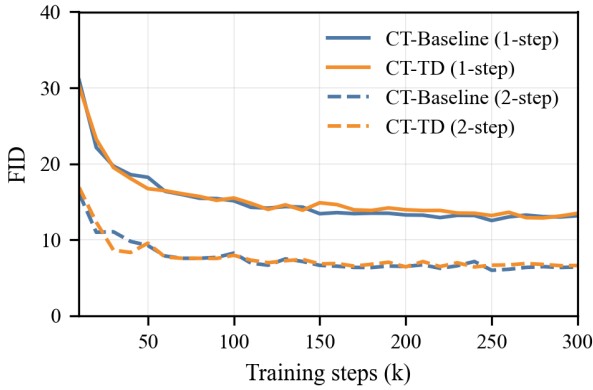

*(a)* AFHQv2 (64 × 64)

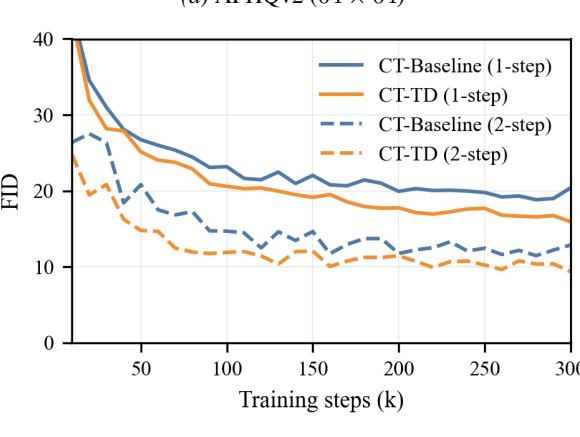

*(b)* FFHQ (64 × 64)

*Figure 2.* Learning curves (FID-50k vs. training steps) for TD+CT and CT baseline. Each plot shows 1-step and 2-step sampling results.

To investigate whether the learned consistency generalizes across various sampling densities, we evaluate the FID difference $\Delta\text{FID}(s) := \text{FID}_{\text{TD+CT}}(s) - \text{FID}_{\text{CT}}(s)$ for step $s \in \{1, 2, 3, 4, 6, 8\}$. As illustrated in Figure 3, although minor fluctuations exist where $\Delta\text{FID}$ occasionally approaches or crosses the zero line (notably for 4 and 8 steps during early or middle training phases), the FID difference remains predominantly negative throughout the majority of the training process across all evaluated step counts. This demonstrates that TD+CT generally maintains a better performance profile over the baseline across different inference budgets.

**Computational Cost.** The additional computational cost of our method mainly arises from maintaining and updating the target network $\theta'$. For TD+EDM, on class-conditional CIFAR-10 with the 55M-parameter UNet trained for 200M images, the baseline training takes 50h 33m 30s (45.7 s/tick), while training with TD takes 71h 36m 48s (64.7 s/tick) on the same hardware. CPU memory increases from 2.29 GB to 2.74 GB (≈20%), and GPU memory from 16.58 GB to 17.51 GB (≈6%). In contrast, for TD+CT, the overhead is

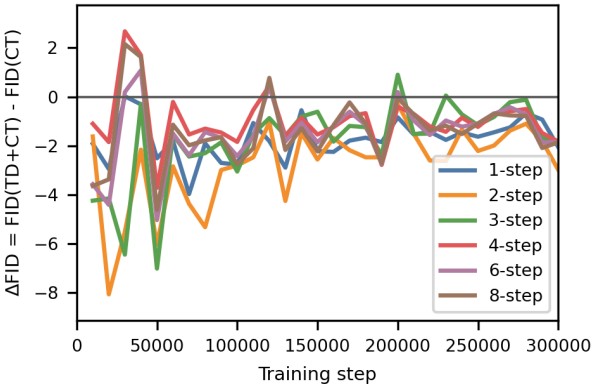

*Figure 3.* Relative improvement of TD+CT over CT baseline on FFHQ ($64 \times 64$) across different inference steps. Negative values indicate that TD+CT achieves lower FID than the baseline.

*Table 4.* Effects of pairwise weighting and sampling steps (CIFAR-10, small UNet). FID-50k ↓ (last-15% avg.; mean ± std over 3 seeds).

| weighted | steps=12 | steps=15 | steps=18 |
|---|---|---|---|
| None | $11.059 \pm 0.110$ | $10.589 \pm 0.155$ | $10.435 \pm 0.183$ |
| $w_{\text{TD}}$ (30) | $\mathbf{10.224 \pm 0.194}$ | $\mathbf{9.755 \pm 0.107}$ | $\mathbf{9.751 \pm 0.093}$ |
| EDM | $10.576 \pm 0.109$ | $10.201 \pm 0.056$ | $9.978 \pm 0.042$ |

minimal because CT itself already utilizes a target network. For 5,000 training steps, TD+CT takes 95min (AFHQv2) and 92min (FFHQ), compared to 80min (AFHQv2) and 78min (FFHQ) for the CT baseline, respectively. These indicate that our method can effectively improve generation quality without incurring significant overhead.

### 4.2. Ablation Study

Given compute constraints, all ablations use CIFAR-10 with a smaller UNet and report last-15% FID-50k (mean ± std over 3 seeds; App. A.3) using TD+EDM. Unless a factor is being swept, the default hyperparameters are $\Delta{=}0.25$, $k{=}1$, $\lambda{=}0.5$, with the weighted TD in Equation (30).

**Effect of pairwise TD weighting.** We first validate the necessity of our sample-wise (pairwise) weighting $w_{\text{TD}}$ in Equation (30) by comparing it against a constant weight (unweighted TD). Table 4 shows that weighting yields a clear improvement across step budgets.

**Sensitivity to the regularization weight $\lambda$.** To study the balance between TD and EDM losses, we sweep $\lambda$ while keeping $\Delta{=}0.25$, $k{=}1$, and weighted TD fixed. Table 5 shows that the best performance is achieved with relatively small $\lambda$, and performance is stable across a low-$\lambda$ range. Moreover, the improvement over EDM is consistent.

**Sensitivity to the stride $\Delta$.** We next investigate sensitivity

to the stride $\Delta$ introduced by our TD formulation. Table 6 shows that, under a fixed step budget, varying $\Delta$ typically leads to only minor differences, suggesting robustness to this hyperparameter.

*Table 5.* Ablation on weight $\lambda$. We vary $\lambda \in \{0.01, 0.5, 1.0, 2.0\}$ under $\Delta{=}0.25$, $k{=}1$. CIFAR-10 FID-50k ↓ (last-15% avg.; mean ± std over 3 seeds).

| $\lambda$ | steps=12 | steps=15 | steps=18 |
|---|---|---|---|
| 0.01 | $\mathbf{10.212 \pm 0.067}$ | $9.793 \pm 0.153$ | $\mathbf{9.750 \pm 0.219}$ |
| 0.50 | $10.224 \pm 0.194$ | $\mathbf{9.755 \pm 0.107}$ | $9.751 \pm 0.093$ |
| 1.00 | $10.367 \pm 0.117$ | $9.977 \pm 0.141$ | $9.635 \pm 0.170$ |
| 2.00 | $10.413 \pm 0.144$ | $9.994 \pm 0.078$ | $9.839 \pm 0.103$ |
| EDM | $10.576 \pm 0.109$ | $10.201 \pm 0.056$ | $9.978 \pm 0.042$ |

*Table 6.* Ablation on stride $\Delta \in \{1/2, 1/3, 1/4, 1/5\}$ (weighted). CIFAR-10 FID-50k ↓ (last-15% avg.; mean ± std over 3 seeds).

| $\Delta$ | steps=12 | steps=15 | steps=18 |
|---|---|---|---|
| 1/2 | $10.331 \pm 0.183$ | $9.698 \pm 0.110$ | $\mathbf{9.688 \pm 0.082}$ |
| 1/3 | $10.264 \pm 0.106$ | $\mathbf{9.634 \pm 0.110}$ | $9.800 \pm 0.069$ |
| 1/4 | $\mathbf{10.224 \pm 0.194}$ | $9.755 \pm 0.107$ | $9.751 \pm 0.093$ |
| 1/5 | $10.232 \pm 0.056$ | $9.787 \pm 0.126$ | $9.706 \pm 0.054$ |
| EDM | $10.576 \pm 0.109$ | $10.201 \pm 0.056$ | $9.978 \pm 0.042$ |

## 5. Discussions

We have introduced a temporal difference (TD) learning framework for diffusion models that reformulates denoising across the time axis as a policy evaluation problem. By casting the denoising trajectory as a Markov reward process (MRP), we derive a novel TD objective that explicitly penalizes inconsistencies in the model's multi-step progress, thereby enforcing cross-time agreement. A key theoretical and practical element of our work is the derivation of a principled, sample-based loss reweighting scheme, $w_{\text{TD}}$, which ensures stable optimization by balancing the loss scales across heterogeneous time pairs.

**Insights from Empirical Results.** Our evaluations with TD+EDM and TD+CT provide empirical evidence for the benefits of temporal consistency training:

- **FID Improvements:** Across multiple standard benchmarks, including CIFAR-10, AFHQv2, and FFHQ, our TD objective generally matches or improves sample quality. For TD+EDM, we observe clear FID reductions in the 12–18 step range on most settings, demonstrating that enforcing cross-time agreement can suppress discretization error accumulation in few-step regimes. For TD+CT, we also observe improved one-step performance on both AFHQv2 and FFHQ, with a particularly large gain on FFHQ (Table 3).
- **Robustness Across Sampling Budgets:** A significant finding is the robustness of the TD+CT model across a

wide range of inference steps. As shown in Figure 3, the relative improvement ($\Delta$FID) remains predominantly negative across all evaluated step counts (1–8 steps), indicating a broadly improved performance profile that is not tied to a single inference budget.

- **Reliability of Hyperparameter Choices:** Our ablation studies demonstrate that the TD framework is robust across various configurations. While the pairwise weighting $w_{\text{TD}}$ is essential for achieving substantial and stable gains, the performance remains consistently superior over the baseline across a broad range of regularization weights $\lambda$ and temporal strides $\Delta$. This confirms the reliability of our proposed default training scheme.

**Limitations.** While effective, our approach has several limitations. First, compared with EDM (Karras et al., 2022), the TD objective may introduce additional constant-factor $\sim 1.5\times$ computational and memory overhead when a separate TD target network is maintained and updated. However, this overhead is largely amortized in training pipelines that already rely on a target/teacher network (e.g., consistency training with EMA teachers), where our method can reuse the existing target network with little marginal cost. Second, the current empirical scope is primarily focused on the generation at resolutions up to $64 \times 64$ using a probability-flow ODE solver. While this setting is consistent with prior algorithmic studies (Ho et al., 2020; Karras et al., 2022), validation on higher-resolution models, broader architectures, and specialized one-/few-step generators remains an important direction.

**Future work.** These findings motivate several promising research directions. The computational cost could be mitigated by amortizing target-network updates or employing lightweight adapter modules. The framework also invites adaptive training curricula, such as progressively widening the step span $k$ or annealing the mixing coefficient $\lambda$ during training (Karras et al., 2022). Related approaches, such as SDPO, have demonstrated the value of dense stepwise rewards (Zhang et al., 2024), and noise-correlation methods like ARTDiff (Lu et al., 2024) could be combined with TD learning to further enhance temporal stability. Finally, given its formulation as a drop-in objective, we anticipate strong synergies with distillation techniques (Salimans & Ho, 2022) and solver-aware training policies (Lu et al., 2022; Zhao et al., 2023).

## Impact Statement

This paper presents work whose goal is to advance the field of Machine Learning. There are many potential societal consequences of our work, none which we feel must be specifically highlighted here.

## Acknowledgement

Qizhen Ying and Yangchen Pan acknowledge the support from the Engineering and Physical Sciences Research Council (EPSRC) New Investigator Award under grant reference UKRI2775. Junfeng Wen acknowledges the support of NSERC, RGPIN-2024-05357. Qizhen Ying acknowledges the use of resources provided by the Isambard-AI National AI Research Resource (AIRR). Isambard-AI is operated by the University of Bristol and is funded by the UK Government's Department for Science, Innovation and Technology (DSIT) via UK Research and Innovation; and the Science and Technology Facilities Council [ST/AIRR/I-A-I/1023] (McIntosh-Smith et al., 2024).

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

*Table 7.* TD+EDM: shared EDM constants.

| Item | Value |
|------|-------|
| $\sigma_{\mathrm{data}}$ (preconditioning) | 0.5 |
| Log-normal noise sampling $(P_{\mathrm{mean}}, P_{\mathrm{std}})$ | $-1.2$, 1.2 |
| Sampling grid $(\sigma_{\min}, \sigma_{\max}, \rho)$ | $2 \times 10^{-3}$, 80, 7 |
| Loss weight $w(\sigma)$ | EDM default |
| Solver / sampler | Probability-flow ODE + Heun |

*Table 8.* TD+EDM: TD regularizer settings.

| Item | Value |
|------|-------|
| Stride $\Delta$ (index unit) | 0.25 |
| Span $k$ | 1.0 |
| Mixing $\lambda$ (Eq. 36) | 0.5 |
| Pairwise weight $w_{\mathrm{TD}}$ (Eq. 30) | on |
| Coupling | Markov increments |
| EMA teacher decay & freq. | 0.999; every step |
| Boundary masking window | $i \in [0, \, N{-}1{-}k{-}\Delta]$ |

# A. Implementation Details

This section provides full implementation settings for all experiments in Sec. 4. We follow the same order as the main text: TD+EDM (Table 2), TD+CT (Table 3, Fig. 2–3), and the ablation study (Tables 4–6).

## A.1. TD+EDM: Full Settings (Table 2)

**Common EDM setup.** Unless stated otherwise, TD+EDM follows the standard EDM configuration (Karras et al., 2022) (training pipeline, EDM preconditioning, noise sampling distribution, and the probability-flow ODE sampler with Heun's method). We summarize the shared EDM constants in Table 7.

**Datasets & preprocessing.** Official train/test split; images normalized to $[-1, 1]$; AugmentPipe with $p = 0.12$ (xflip, yflip, scale, rotate, aniso, translate), following the official PyTorch implementation of Karras et al. (2022). In Table 2, CIFAR-10 experiments are **class-conditional**, while FFHQ and AFHQv2 are **unconditional**.

**Model architecture (EDM default setting).** We use the EDM default SongUNet (DDPM++) backbone with EDM preconditioning and $\hat{x}_0$ parameterization.

**TD objective settings (ours).** Default TD hyperparameters: stride $\Delta = 0.25$ (index unit), span $k = 1$, mixing $\lambda = 0.5$, and the TD pairwise weighting proposed in Sec. 3.3. Coupling: Markov increments. EMA target network decay $\tau = 0.999$, updated every step. Boundary mask near $\sigma_{\min}$ with valid index range $i \in [0, \, N - 1 - k - \Delta]$. Pair sampling: uniform over valid indices.

**Optimization and schedule (EDM default setting).** Adam, learning rate $2 \times 10^{-4}$, betas $(0.9, 0.999)$; no weight decay, no grad clip; constant LR, no warmup; batch 128 per GPU, global 512, no accumulation; precision fp32 (static scale 1); training length 200M images ($\sim$4000 ticks) for all datasets in Table 2.

**Sampler & inference (EDM default setting).** Probability-flow ODE; Heun (order 2). We evaluate inference steps among $\{9, 12, 15, 18\}$, and report $NFE = 2 \times$ steps $- 1$ (i.e., NFE $\in \{17, 23, 29, 35\}$). We use the same $\rho$-power grid $(\sigma_{\min}, \sigma_{\max}, \rho) = (2 \times 10^{-3}, 80, 7)$.

**Evaluation protocol.** FID-50k with deterministic evaluation seeds 0–49999; we report the **last-15%** average over evaluation checkpoints (as described in Sec. 4.1).

**Computational resources:** $4 \times$H100 80GB

**A.2. TD+CT Experimental Details (Table 3)**

**Experiment setup.** TD+CT follows the consistency training (CT) setup of Song et al. (2023) for each dataset (architecture, parameterization, schedules, solver, and evaluation protocol), unless explicitly stated otherwise.

**Training and hyperparameters.** For both datasets, we use the **adaptive** $N$ schedule in Song et al. (2023), where the number of discretization steps $N(\cdot)$ increases during training. Models are trained for a total of **300,000 iterations**. We sweep the TD-specific hyperparameters $\lambda$ and $\Delta$ for TD+CT, and select $\Delta = 0.5$, $k = 1$, and $\lambda = 0.5$ based on the last-15% one-step FID-50k, together with the CT-specific pairwise TD weight and the same boundary masking rule as in Sec. 3.3.

**Evaluation.** We evaluate every 5,000 training iterations. We report one-step FID-50k (steps= 1, NFE= 1) using a single Heun update and plot the resulting learning curves in Fig. 2. For multi-step analysis (Fig. 3), we also evaluate $s \in \{1, 2, 3, 4, 6, 8\}$ at each evaluation point and report $\Delta\text{FID}(s)$ as defined in the main text.

**Computational resources:** $4\times$H100 80GB

**A.3. Small-model Ablations (Tables 4–6)**

**Scope.** All ablations use CIFAR-10 with a smaller UNet and report last-15% FID-50k (mean $\pm$ std over 3 seeds), as described in Sec. 4.2. Unless a factor is being swept, we anchor hyperparameters at $\Delta = 0.25$, $k = 1$, $\lambda = 0.5$ with the *weighted* TD in Eq. 30.

**Datasets & preprocessing.** CIFAR-10; official split; resolution $32\times32$ for train/eval; normalized to $[-1, 1]$; AugmentPipe with $p = 0.12$ (xflip, yflip, scale, rotate, aniso, translate). **Notice:** in contrast to Table 2 (conditional CIFAR-10 for TD+EDM), all ablations on CIFAR-10 are unconditional.

**Model architecture.** SongUNet (DDPM++), $\sim$13M parameters; base channels $ch$=64, multipliers $[1, 2, 2]$; 2 res-blocks/level; attention at 16 with 1 head, head_dim 64; GroupNorm + SiLU; dropout 0.10; time/noise embed dims 256 (positional); output parameterization $\hat{x}_0$ (EDM).

**EDM constants & TD settings.** We use the same EDM constants as in Table 7. Unless swept, we use the same TD settings as in Table 8. The ablated ranges of $\Delta$, $\lambda$, and weighting are summarized in Table 9.

**Optimization and training length.** Follow the EDM default optimizer setup (Table 2) with Adam (LR $2\times10^{-4}$, betas $(0.9, 0.999)$), no weight decay, no grad clip, constant LR, no warmup. We train each configuration for 120M images ($\sim$2400 ticks).

**Sampler & evaluation.** Probability-flow ODE + Heun (order 2). We evaluate steps among $\{12, 15, 18\}$ and report $NFE = 2 \times$ steps $- 1$ (i.e., NFE $\in \{23, 29, 35\}$). FID-50k with deterministic evaluation seeds 0–49999; last-15% average.

**Computational resources:** single NVIDIA RTX4090.

*Table 9.* Ablation ranges for TD hyperparameters on CIFAR-10 (small UNet).

| Item | Value |
|------|-------|
| "One-step" stride (index unit) | $1/2, 1/3, 1/4, 1/5$ |
| "$k$-step" span $k$ | $1.0$ |
| Mixing $\lambda$ (Eq. 36) | $0.01, 0.5, 1.0, 2.0$ |
| Pairwise weight $w_{\text{TD}}$ (Eq. 30) | weighted, unweighted |
| EMA teacher decay & freq. | $0.999$; every step |
| Boundary masking window | $i \in [0, N-1-k-\Delta]$ |

# B. Forward Processes and Two-Time Means

This appendix consolidates the forward/noising processes considered in the paper and rewrites their two-time posterior means in the unified linear form

$$\boldsymbol{\mu}_\tau^{\text{true}}(\boldsymbol{x}_t, \boldsymbol{x}_0) = A_{t,\tau}\,\boldsymbol{x}_0 + \kappa_{t,\tau}\,\boldsymbol{x}_t, \qquad \tau < t, \tag{39}$$

where $x_0 \in \mathbb{R}^d$ is the clean datum and $x_t$ is a noisy observation at level $t$ on the *same* time/noise axis as $\tau$. For discrete-time models (DDPM/DDIM), $t, \tau \in \{0, \ldots, T\}$ and we typically take $\tau = t-1$ on the native grid. For continuous-time models (VP/VE/EDM), $t, \tau \in [0, T]$ (or a monotone reparameterization such as $\sigma$) chosen by the sampler.

**Notation.** For DDPM/DDIM, set $\alpha_t = 1 - \beta_t$ and $\bar{\alpha}_t = \prod_{s=1}^{t} \alpha_s$. For VP-SDE, let $\alpha(t) = \exp\left(-\frac{1}{2}\int_0^t \beta(s)\, ds\right)$ and $\sigma(t) = \sqrt{1 - \alpha(t)^2}$. For VE/EDM, $\sigma(\cdot)$ denotes the noise scale with $x_t = x_0 + \sigma(t)\,\epsilon$, $\epsilon \sim \mathcal{N}(0, I)$. All random vectors are in $\mathbb{R}^d$, and $I$ is the identity matrix.

## B.1. DDPM (discrete time)

**Forward one-step transition.** The DDPM forward transition and the marginal w.r.t. $x_0$ are (Luo (2022), Eq.(31); Eq.(69)-(70))):

$$q(x_t \mid x_{t-1}) = \mathcal{N}\left(\sqrt{\alpha_t}\, x_{t-1},\ (1 - \alpha_t)I\right), \quad \text{(DDPM-Forward)} \tag{40}$$

$$q(x_t \mid x_0) = \mathcal{N}\left(\sqrt{\bar{\alpha}_t}\, x_0,\ (1 - \bar{\alpha}_t)I\right), \quad \text{(DDPM-Marginal)} \tag{41}$$

with the equivalent reparameterization

$$x_t = \sqrt{\bar{\alpha}_t}\, x_0 + \sqrt{1 - \bar{\alpha}_t}\, \epsilon, \quad \epsilon \sim \mathcal{N}(0, I). \quad \text{(DDPM-Reparam)} \tag{42}$$

**Posterior toward $t-1$ (mean and variance).** By linear-Gaussian conditioning,

$$q(x_{t-1} \mid x_t, x_0) = \mathcal{N}\left(\mu_{t-1}^{\text{true}}(x_t, x_0),\ \tilde{\beta}_t I\right), \quad \text{(DDPM-Posterior)} \tag{43}$$

$$\text{where } \mu_{t-1}^{\text{true}}(x_t, x_0) = \frac{\sqrt{\bar{\alpha}_{t-1}}\,(1 - \alpha_t)}{1 - \bar{\alpha}_t}\, x_0 + \frac{\sqrt{\alpha_t}\,(1 - \bar{\alpha}_{t-1})}{1 - \bar{\alpha}_t}\, x_t, \quad \text{(DDPM-PosteriorMean)} \tag{44}$$

$$\tilde{\beta}_t = \frac{(1 - \alpha_t)(1 - \bar{\alpha}_{t-1})}{1 - \bar{\alpha}_t}. \quad \text{(DDPM-PosteriorVar)} \tag{45}$$

Equations (44)-(45) from Luo (2022, Eq.(84)-(85), p.12).

**SNR identity.** We will also refer to the signal-to-noise ratio

$$\text{SNR}(t) = \frac{\bar{\alpha}_t}{1 - \bar{\alpha}_t}, \quad \text{(DDPM-SNR)} \tag{46}$$

as used to simplify weighting expressions (Luo (2022), Eq.(109), p.14)).

**Two-time mean in unified form.** Taking $\tau = t-1$, the DDPM two-time mean $\mu_{t-1}^{\text{true}}(x_t, x_0)$ equals Equation (44), i.e., the unified form equation (39) with

$$A_{t,\tau} = \frac{\sqrt{\bar{\alpha}_{t-1}}\,(1 - \alpha_t)}{1 - \bar{\alpha}_t}, \qquad \kappa_{t,\tau} = \frac{\sqrt{\alpha_t}\,(1 - \bar{\alpha}_{t-1})}{1 - \bar{\alpha}_t}.$$

## B.2. DDIM: Two-Time Mean and Parameters (short)

Use the same notation as DDPM, let $\alpha_t = 1 - \beta_t$ and $\bar{\alpha}_t = \prod_{s=1}^{t} \alpha_s$. DDIM specifies a non-Markovian reverse conditional whose mean depends on $(x_t, x_0)$ Song et al. (2021a, Eq. 7)):

$$q_\sigma(x_{t-1} \mid x_t, x_0) = \mathcal{N}\left(\underbrace{\sqrt{\bar{\alpha}_{t-1}}\, x_0 + \sqrt{\frac{1 - \bar{\alpha}_{t-1} - \sigma_t^2}{1 - \bar{\alpha}_t}}\,\left(x_t - \sqrt{\bar{\alpha}_t}\, x_0\right)}_{\text{mean}},\ \sigma_t^2 I\right). \tag{47}$$

Hence in our unified linear form, we have

$$\kappa_{t,t-1} = \sqrt{\frac{1 - \bar{\alpha}_{t-1} - \sigma_t^2}{1 - \bar{\alpha}_t}}, \qquad A_{t,t-1} = \sqrt{\bar{\alpha}_{t-1}} - \kappa_{t,t-1}\sqrt{\bar{\alpha}_t}. \tag{48}$$

**Stochasticity and the $\eta$-parameterization (DDIM Eq. (16)).** A convenient schedule for the reverse variance is

$$\sigma_t(\eta) = \eta \sqrt{\frac{1 - \bar{\alpha}_{t-1}}{1 - \bar{\alpha}_t}} \sqrt{1 - \frac{\bar{\alpha}_t}{\bar{\alpha}_{t-1}}}, \qquad \eta \in [0, 1]. \tag{49}$$

Special cases: (i) $\eta=0 \Rightarrow \sigma_t=0$ gives the deterministic DDIM, where $\kappa_{t,t-1} = \sqrt{\frac{1-\bar{\alpha}_{t-1}}{1-\bar{\alpha}_t}}$ and $A_{t,t-1} = \sqrt{\bar{\alpha}_{t-1}} - \sqrt{\frac{1-\bar{\alpha}_{t-1}}{1-\bar{\alpha}_t}} \sqrt{\bar{\alpha}_t}$; (ii) $\eta=1$ recovers the DDPM variance choice at step $t$.

## B.3. VP–SDE (variance preserving)

**Forward dynamics and one-time marginals.** The VP SDE is

$$d\boldsymbol{x}_t = -\tfrac{1}{2} \beta(t) \, \boldsymbol{x}_t \, dt + \sqrt{\beta(t)} \, d\boldsymbol{w}_t, \qquad t \in [0, 1], \tag{VP-SDE}$$

as stated in Eq. (11) of Song et al. (2021b). Let

$$\alpha(t) := \exp\left(-\tfrac{1}{2} \int_0^t \beta(s) \, ds\right), \qquad \sigma(t) := \sqrt{1 - \alpha(t)^2}. \tag{50}$$

Solving Equation (VP-SDE) gives the Gaussian marginal

$$p_{0t}(\boldsymbol{x}_t \mid \boldsymbol{x}_0) = \mathcal{N}\big(\alpha(t) \, \boldsymbol{x}_0, \, (1 - \alpha(t)^2)I\big), \tag{VP-kernel}$$

which appears as the VP case of Eq. (29) in Song et al. (2021b).

**Two-time conditional mean.** Fix $0 \leq \tau < t \leq 1$. From the linear solution of equation (VP-SDE), we can write

$$\boldsymbol{x}_t = \alpha(t) \, \boldsymbol{x}_0 + \alpha(t) \int_0^t \frac{\sqrt{\beta(s)}}{\alpha(s)} \, d\boldsymbol{w}_s, \qquad \boldsymbol{x}_\tau = \alpha(\tau) \, \boldsymbol{x}_0 + \alpha(\tau) \int_0^\tau \frac{\sqrt{\beta(s)}}{\alpha(s)} \, d\boldsymbol{w}_s.$$

Conditioned on $\boldsymbol{x}_0$, $(\boldsymbol{x}_\tau, \boldsymbol{x}_t)$ is jointly Gaussian with

$$\begin{aligned}
&\mathbb{E}[\boldsymbol{x}_\tau \mid \boldsymbol{x}_0] = \alpha(\tau)\boldsymbol{x}_0, \quad \mathbb{E}[\boldsymbol{x}_t \mid \boldsymbol{x}_0] = \alpha(t)\boldsymbol{x}_0, \\
&\mathrm{Cov}(\boldsymbol{x}_\tau \mid \boldsymbol{x}_0) = (1 - \alpha(\tau)^2)I, \quad \mathrm{Cov}(\boldsymbol{x}_t \mid \boldsymbol{x}_0) = (1 - \alpha(t)^2)I, \\
&\mathrm{Cov}(\boldsymbol{x}_\tau, \boldsymbol{x}_t \mid \boldsymbol{x}_0) = \frac{\alpha(t)}{\alpha(\tau)}\big(1 - \alpha(\tau)^2\big)I,
\end{aligned}$$

where cross-covariance follows Itô isometry: $\mathrm{Cov}(\boldsymbol{x}_\tau, \boldsymbol{x}_t \mid \boldsymbol{x}_0) = \alpha(t)\alpha(\tau) \int_0^\tau \alpha(s)^{-2}\beta(s) \, ds = \frac{\alpha(t)}{\alpha(\tau)}\big(1 - \alpha(\tau)^2\big)I$. Applying the Gaussian conditioning formula—for jointly Gaussian vectors $(y_1, y_2)$, the conditional mean is $\mathbb{E}[y_1 \mid y_2] = \boldsymbol{\mu}_1 + \Sigma_{12}\Sigma_{22}^{-1}(y_2 - \boldsymbol{\mu}_2)$—then yields

$$\boldsymbol{\mu}_\tau^{\text{true}}(\boldsymbol{x}_t, \boldsymbol{x}_0) = \alpha(\tau) \, \boldsymbol{x}_0 + \underbrace{\left[\frac{\alpha(t)}{\alpha(\tau)} \cdot \frac{1 - \alpha(\tau)^2}{1 - \alpha(t)^2}\right]}_{\kappa_{t,\tau}} \big(\boldsymbol{x}_t - \alpha(t) \, \boldsymbol{x}_0\big). \tag{VP-TwoTime}$$

Equivalently, in the unified linear form $\boldsymbol{\mu}_\tau^{\text{true}}(\boldsymbol{x}_t, \boldsymbol{x}_0) = A_{t,\tau}x_0 + \kappa_{t,\tau}x_t$,

$$A_{t,\tau} = \alpha(\tau) - \kappa_{t,\tau} \, \alpha(t), \qquad \kappa_{t,\tau} = \frac{\alpha(t)}{\alpha(\tau)} \cdot \frac{1 - \alpha(\tau)^2}{1 - \alpha(t)^2}. \tag{51}$$

*Remark.* With the commonly used linear schedule $\beta(t) = \bar{\beta}_{\min} + t(\bar{\beta}_{\max} - \bar{\beta}_{\min})$, the corresponding $p_{0t}(\boldsymbol{x}_t \mid \boldsymbol{x}_0)$ is given explicitly in Eqs. (32)–(33) of Song et al. (2021b).

### B.4. VE-SDE and EDM (variance exploding & $\sigma$-parameterization)

**Forward dynamics and one-time marginals.** The VE SDE reads

$$\mathrm{d}\boldsymbol{x}_t = \sqrt{\tfrac{d}{dt}\,\sigma(t)^2}\ \mathrm{d}\boldsymbol{w}_t, \qquad \sigma(0) = 0, \tag{VE-SDE}$$

see Eq. (9) in Song et al. (2021b). It induces the additive-noise marginal

$$p_{0t}(\boldsymbol{x}_t \mid \boldsymbol{x}_0) = \mathcal{N}\big(\boldsymbol{x}_0,\ \sigma(t)^2 I\big), \tag{VE-kernel}$$

the VE case of Eq. (29) in Song et al. (2021b). $\boldsymbol{x}_t = \boldsymbol{x}_\tau + \sqrt{\sigma(t)^2 - \sigma(\tau)^2}\,\boldsymbol{z}$ with $\boldsymbol{z} \sim \mathcal{N}(\boldsymbol{0}, I)$ independent of $\boldsymbol{x}_\tau$.

**Two-time conditional mean for VE.** With the above coupling, conditioned on $\boldsymbol{x}_0$,

$$\mathrm{Cov}(\boldsymbol{x}_t \mid \boldsymbol{x}_0) = \sigma(t)^2 I, \qquad \mathrm{Cov}(\boldsymbol{x}_\tau \mid \boldsymbol{x}_0) = \sigma(\tau)^2 I, \qquad \mathrm{Cov}(\boldsymbol{x}_\tau, \boldsymbol{x}_t \mid \boldsymbol{x}_0) = \sigma(\tau)^2 I.$$

Thus for $\tau > 0$ and $t > \tau$,

$$\boldsymbol{\mu}_\tau^{\mathrm{true}}(\boldsymbol{x}_t, \boldsymbol{x}_0) = \boldsymbol{x}_0 + \underbrace{\frac{\sigma(\tau)^2}{\sigma(t)^2}}_{\kappa_{t,\tau}}(\boldsymbol{x}_t - \boldsymbol{x}_0) = \left(1 - \frac{\sigma(\tau)^2}{\sigma(t)^2}\right)\boldsymbol{x}_0 + \frac{\sigma(\tau)^2}{\sigma(t)^2}\,\boldsymbol{x}_t. \tag{VE-TwoTime}$$

Hence the unified coefficients are $A_{t,\tau} = 1 - \frac{\sigma(\tau)^2}{\sigma(t)^2}$ and $\kappa_{t,\tau} = \frac{\sigma(\tau)^2}{\sigma(t)^2}$.

**EDM uses the same corruption and thus the same two-time mean.** EDM (Karras et al., 2022) parameterizes "time" directly by the noise scale $\sigma$ and corrupts clean data additively: $\boldsymbol{x} = \boldsymbol{x}_0 + \boldsymbol{n}$, $\boldsymbol{n} \sim \mathcal{N}(\boldsymbol{0}, \sigma^2 I)$. Under the same Markov-increments coupling as in VE, the joint $(\boldsymbol{x}_\tau, \boldsymbol{x}_t) \mid \boldsymbol{x}_0$ is Gaussian with the same covariances as above, so the two-time mean of EDM coincides with VE's formula.

**Shared-noise coupling for CT.** Consistency training uses the same noise across times:

$$\boldsymbol{x}_\tau = \boldsymbol{x}_0 + \sigma(\tau)\,\boldsymbol{\epsilon}, \qquad \boldsymbol{x}_t = \boldsymbol{x}_0 + \sigma(t)\,\boldsymbol{\epsilon}, \qquad \boldsymbol{\epsilon} \sim \mathcal{N}(\boldsymbol{0}, I).$$

In this case $\boldsymbol{x}_\tau - \boldsymbol{x}_0 = \frac{\sigma(\tau)}{\sigma(t)}(\boldsymbol{x}_t - \boldsymbol{x}_0)$ deterministically, so the posterior collapses and the mean is

$$\boldsymbol{\mu}_\tau^{\mathrm{true}}(\boldsymbol{x}_t, \boldsymbol{x}_0) = \boldsymbol{x}_0 + \underbrace{\frac{\sigma(\tau)}{\sigma(t)}}_{\kappa_{t,\tau}}(\boldsymbol{x}_t - \boldsymbol{x}_0) = \left(1 - \frac{\sigma(\tau)}{\sigma(t)}\right)\boldsymbol{x}_0 + \frac{\sigma(\tau)}{\sigma(t)}\boldsymbol{x}_t. \tag{CT-TwoTime}$$

Hence $A_{t,\tau} = 1 - \frac{\sigma(\tau)}{\sigma(t)}$ and $\kappa_{t,\tau} = \frac{\sigma(\tau)}{\sigma(t)}$.

