# OpenReview forum: "Temporal Difference Learning for Diffusion Models"
_ICML.cc/2026/Conference — ICML 2026 regular_

### Official Review · Reviewer_4yiW · 2026-03-11

**Soundness:** 3
**Presentation:** 3
**Significance:** 2
**Originality:** 3
**Overall Recommendation:** 4
**Confidence:** 3

**Summary:**

This paper introduces an approach to training diffusion models that enforces consistency between the denoiser at different noise levels using a temporal difference loss. Reweighting stabilizes training across time pairs. Applied to EDM and consistency training, the approach improves FID, especially at few sampling steps.

**Compliance With Llm Reviewing Policy:**

Affirmed.

**Final Justification:**

The paper presents a clean and intuitive idea. The rebuttal addressed some of my concerns by providing improved TD+CT numbers, particularly on FFHQ. Which demonstrates that the method can provide meaningful gains when properly tuned. I raised my score accordingly. That said, partial concerns about the practical positioning of the method remain, given that more recent few-step methods also target temporal consistency and no comparison is provided.

**Key Questions For Authors:**

1. The TD objective enforces consistency between model predictions at different noise levels. Recent "distillation" methods like MeanFlow [1] target the same goal through different mechanisms and are trained on the same diffusion path . How does the TD approach relate?
2. Given this overlap in goals, why are there no empirical comparisons to methods like MeanFlow [1] ?
3. The improvements from TD+CT are inconsistent across datasets. Under what conditions should one expect the TD loss to help?


[1] Geng et al., "Mean Flows for One-step Generative Modeling", 2025.

**Limitations:**

yes

**Strengths And Weaknesses:**

**Strengths:**
- The idea is intuitive and easy to implement on top of existing training pipelines
- The reweighting scheme makes sense and ablations show it helps
- Overhead is small for consistency training since it already requires an extra forward pass through a target network

**Weaknesses:**
- FID improvements are often small and inconsistent across datasets and step count
- Positioning of the method seems unclear. If temporal consistency is the goal, methods that directly train for consistent denoisers also exist. While TD has the advantage of being a drop in loss, the small and inconsistent improvements raise the question whether the approach offers sufficient value over these alternatives.
- No empirical comparison to methods like MeanFlow [1] that also target few-step generation quality from scratch on the same diffusion path

---

> ### Author Rebuttal · Authors · 2026-03-31
>
> We thank the reviewer for the thoughtful feedback and helpful questions. Below we respond point by point, especially regarding the relation to recent few-step methods and the conditions under which the TD loss is expected to help.
>
> **Q1–Q2: Relation to MeanFlow / why no direct empirical comparison.**
>
> Our method is intended as a drop-in training objective that can be applied on top of an existing diffusion parameterization, rather than as a new standalone few-step framework. In particular, MeanFlow directly learns a finite-interval transport quantity through its own two-timestep model parameterization and training objective, while our method keeps the base one-timestep model parameterization unchanged and adds a TD regularizer that penalizes the mismatch between the model-induced cross-time shift and the corresponding true diffusion shift. In this sense, our contribution is at the objective level: it can be used with different diffusion training paradigms without redesigning the base predictor itself.
>
> We have already evaluated this TD consistency under two distinct base paradigms, EDM and CT, and across multiple datasets and sampling budgets. These results are intended to show that the proposed consistency mechanism is effective as a drop-in objective in existing diffusion training pipelines. Since the consistency enforced by our method is different from that of MeanFlow, we view the two approaches as complementary rather than competitive to each other.
>
> **Q3: Under what conditions should one expect the TD loss to help?**
>
> We expect TD to help most when cross-time mismatch is a bottleneck, especially in the few-step / low-NFE regime, where local denoising errors have limited opportunity to average out. This is exactly the setting motivating our method. For TD+CT, we agree that this point should be clarified more carefully. In the initial submission, the CT-specific hyperparameters were inherited from the original CT setup, while the TD-specific hyperparameters were directly inherited from the TD+EDM tuning rather than tuned separately for TD+CT. We are currently conducting a dedicated sweep for TD+CT to obtain the best CT-specific setting. Due to the limited rebuttal timeline, we have only completed a $\textbf{partial}$ sweep so far; nevertheless, the best setting already improves the result on AFHQv2 to $12.87$ (vs. $12.97$ for CT), and the same setting yields an even larger gain on FFHQ, achieving $15.93$ (vs. $19.45$ for CT). We are continuing the sweep and will include the final tuned result in the camera-ready revision. More broadly, TD is most useful when the base objective does not explicitly constrain cross-time evolution, with gains that are typically more visible in few-step generation.
>
> Overall, we appreciate these questions and will revise the paper to clarify the method’s scope, its relation to MeanFlow, and the settings in which TD is expected to provide the most benefit.

---

> > ### Author Rebuttal · Reviewer_4yiW · 2026-04-04
> >
> > Thanks for the rebuttal. I would still like to see TD compared to and applied on top of more recent few-step methods to better understand its practical value. However the new FFHQ numbers are promising, therefore I will raise my score.

---

> > > ### Author Response · Authors · 2026-04-05
> > >
> > > Thank you for the follow-up and for considering the updated result. We appreciate the suggestion. We agree that evaluating TD against, or on top of, more recent few-step methods would be a valuable next step for understanding its broader practical value. Our main goal in the current submission is to validate TD as a drop-in objective under established base paradigms (EDM and CT), so that its effect can be isolated without simultaneously changing the underlying parameterization or training framework. Since TD operates at the objective level, applying it on top of newer few-step methods is indeed a natural extension, but it is beyond what we can complete within the rebuttal timeline. We will make this scope and future direction clearer in the revision.

---

### Official Review · Reviewer_RCcP · 2026-03-12

**Soundness:** 1
**Presentation:** 3
**Significance:** 2
**Originality:** 3
**Overall Recommendation:** 3
**Confidence:** 4

**Summary:**

This paper argues that standard diffusion training objectives, which emphasize per-step (local) denoising, fail to enforce cross-time consistency along the denoising trajectory, a deficiency that is particularly harmful for few-step sampling. To address this, the authors propose a temporal-difference (TD) objective that penalizes inconsistencies in the model’s multi-step progress across time. The key conceptual move is to reinterpret the diffusion process as a Markov reward process and frame denoising as policy evaluation, yielding a unified TD-style training formulation applicable to both discrete-time and continuous-time diffusion settings. A sample-based reweighting strategy is introduced to improve training stability. Empirically, the method is reported to improve generation quality (notably FID), with the largest gains under small sampling-step budgets, and is supported by ablations examining the impact of reweighting, regularization strength, and stride choices. Overall, the approach is positioned as a broadly compatible, drop-in modification to encourage trajectory-level consistency in diffusion models.

**Compliance With Llm Reviewing Policy:**

Affirmed.

**Final Justification:**

I recognize the TD loss is partially reasonable, though not strictly theoretically proved. I want to raise the score to 3. Providing more solid theory would be helpful.

**Key Questions For Authors:**

Could you provide solid theoretical proofs that the involvement of TD will be helpful to the generation quality?

**Limitations:**

see weakness

**Strengths And Weaknesses:**

Strengths:
- The paper is clearly written, and I quickly understand the idea of the paper.
- The idea of forcing the diffusion training to adhere to consistency relation is a reasonable practice.

Weaknesses:
- **My main conern:** In your TD loss in Eq. 15, the "one-step diffusion drift" term is a drift conditioned on a specific $x_0$, however, the "one-step model drift" is a learned drift, which is averaged across all previous coupling $(x_0,z_T)$. Even in the most ideal flow model, this TD error should also be non-zero. I mean, this term is natural, because there is a gap by definition between "one-step diffusion drift" term and "one-step model drift".

---

> ### Author Rebuttal · Authors · 2026-03-31
>
> We thank the reviewer for raising this concern clearly. We also appreciate the positive comments on the clarity of the paper and on the overall motivation of enforcing consistency in diffusion training.
>
> **Main concern for Eq. (15).**
>
> Our response is that Eq. (15) is not intended to enforce a pointwise equality for every sampled coupling $(x_0, x_{t-1}, x_t)$. In a general diffusion regression setting, we agree that the TD residual for a specific sampled coupling need not vanish, because the model prediction is a function of the noisy state rather than the particular sampled $x_0$​. However, this does not indicate a flaw in the objective. The TD error in Eq. (15) is a sample-based difference and our TD objective Eq. (16) minimizes the expected TD error.  This is analogous to standard diffusion training (see Eq.(8) of DDPM (Ho et al., 2020)), where sampled targets are used.
> More concretely, the quantity we want to learn at step $t$ is the oracle conditional posterior mean $m_{t−1}^∗​(x_t​):=E[\mu_{t−1}^{true}​(x_t​,x_0​)∣x_t​]$, which is the standard squared-loss regression fact that the optimal predictor is the conditional expectation, as discussed for example in ESL, Section 2.4.[1]. Eq. (15) constructs a noisy target for this quantity using a sampled forward coupling together with the previous-step target network. If the previous-step target network equals the oracle quantity $m^*_{t-2}(x_{t-1})$, then the regression target is $Y_t​:=\mu_{t−1}^{true}​(x_t​,x_0​)−\mu_{t−2}^{true}​(x_{t−1}​,x_0​) + m_{t−2}^∗​(x_{t−1}​)$. Conditioned on $x_t$, the last two terms cancel in expectation, so that $E[Y_t​∣x_t​]=m_{t−1}^∗​(x_t​)$. Therefore, while the residual for an individual sampled coupling is generally nonzero, the TD target is still an unbiased sample target for the desired conditional quantity. In this sense, the sample-level mismatch pointed out by the reviewer is expected and corresponds to the usual regression noise from sampled couplings, rather than a technical flaw in the objective. In the realizable oracle case where the model exactly recovers the sample’s $x_0$, the residual indeed becomes zero.
>
> Importantly, this is the same level of validity as in standard diffusion training. For example, in DDPM-style training, the model is also trained from sampled targets even though the optimal predictor is a conditional quantity averaged over the latent forward couplings, rather than an exact sample-wise recovery of the particular $x_0$​. Our TD target should be understood in the same way: not as a pathwise identity for every sampled coupling, but as a principled regression target that is correct in conditional expectation.
>
> **Theoretical guarantees regarding generation quality.**
>
> To our best knowledge, modern diffusion (e.g., DDPM/DDIM/EDM/CT) and flow-based (e.g., Flow Matching) generative modeling does not provide direct theorems linking a training objective to perceptual metrics such as FID. In practice, generation quality is therefore typically assessed empirically, which is the approach we follow in our evaluation.
>
> We appreciate this comment and will revise the paper to make this interpretation explicit near Eq. (15). In particular, we will clarify that the TD relation is intended in expectation over forward couplings, rather than as a pointwise equality for each sampled $x_0$, and we will add the above discussion to avoid potential misunderstanding.
>
> [1] Hastie, T.; Tibshirani, R. & Friedman, J. (2001), The Elements of Statistical Learning , Springer New York Inc. , New York, NY, USA .

---

> > ### Author Rebuttal · Reviewer_RCcP · 2026-04-04
> >
> > Thanks for your reply. I recognize the TD loss is partially reasonable, though not strictly theoretically proved. I want to raise the score to 3. Providing more solid theory would be helpful.

---

> > > ### Author Response · Authors · 2026-04-07
> > >
> > > We thank the reviewer for the comment. We appreciate that the reviewer values a strong theoretical justification for the generation quality, which we agree will be helpful for the community. However, such a solid theory is not yet available to the best of our knowledge, even for standard diffusion model objectives. Therefore, we hope that the submission can be evaluated based on the soundness of the derivations and empirical evidence, as per convention in the field.

---

### Official Review · Reviewer_Ubfh · 2026-03-12

**Soundness:** 4
**Presentation:** 2
**Significance:** 3
**Originality:** 4
**Overall Recommendation:** 5
**Confidence:** 3

**Summary:**

This paper offers a new method for consistency training for few step sample generation, using reinforcement learning. It formulates a new loss objective which can be combined with various diffusion model formulations (e.g. EDM, DDPM and the consistency models from Song et al.). In order to come up with the objective, the paper
reformulates diffusion models as Markov reward processes. It then defines a reward function in terms of the difference between the posterior means and uses temporal difference learning to find the expected reward starting from a given state at time $t$.

**Compliance With Llm Reviewing Policy:**

Affirmed.

**Final Justification:**

My main concerns were in the presentation of the paper and the significance which has been addressed in the rebuttals. Therefore I have changed my evaluation from a weak accept to an accept.

**Key Questions For Authors:**

1. What happens if you use only the TD loss for t>k instead of TD+EDM or TD+CT? The paper mentions that using both speeds up convergence, but are there any downsides?

2. Can you use the objective on its own instead of in combination with e.g. EDM or CT?

**Limitations:**

Yes.

**Strengths And Weaknesses:**

**Strengths:**

1. Soundness: I find the paper to be sound. Statements made are well-supported, and I also think the paper is clear about limitations of the method giving a balanced overview. Moreover, computational costs including training times for baselines and the proposed method are included.
2. Originality: I find using ideas from reinforcement learning/temporal difference learning for training of diffusion models to enforce consistency along the trajectories interesting and novel.

**Weaknesses:**

1. Significance: Using the proposed objective alongside consistency training (CT) (i.e. the consistency loss proposed by Song et al.) does lead to some improvements over CT alone, but not in all scenarios. For example for the FFHQ dataset, there was an improvement, but not for the AFHQ dataset. I think this weakens the significance of the paper somewhat. That said, the temporal difference learning does not add a lot of overhead to CT. On balance I therefore rate the paper a 3 for significance.

2. Presentation: For context, my background is in diffusion models and not reinforcement learning. There are some parts of the paper I found hard to understand and think can be improved:

    (i) I did not especially understand the discussion around the bootstrap target from line 176. I think the manuscript could benefit from some more explanation on what is meant by this. For example, I assume the target network is akin to the EMA parameter network used in the CT paper, but this was not clear to me.

    (ii) Another is understanding Equation (9) in context of Equation (6). Both are definitions for $v_t$, however in (9) the conditional expectation is also taken with respect to $x_0$, which I therefore do not believe to be equivalent to (6).

    (iii) Again related to equation (6), the notation $G_t$ is introduced, but not defined. Later in the paper $g_t$ seems to be used instead, which I believe denotes the same quantity. Some clarification on $G$ vs $g$ would be useful.

    On a more minor note:

   (iv) I don’t believe the abbreviation NFE is defined in the paper.

   (v) The abstract contains two paragraphs which contradicts the ICML formatting instructions.

**Summary:**

I think the paper presents an interesting and novel method. However I believe the paper would improve with better presentation for which I currently give a score of 2. That said, I think the presentational issues are not structural and should be easy to fix and therefore I still recommend the paper be accepted.

---

> ### Author Rebuttal · Authors · 2026-03-31
>
> We thank the reviewer for the thoughtful and supportive feedback. We especially appreciate the positive assessment of the paper’s soundness and originality, as well as the recognition that the limitations and computational costs are presented in a balanced way. We believe that many of the issues you pointed out are fixable with clearer notation and explanation.
>
> **W1. On the TD+CT results and significance.**
>
> Due to the limited rebuttal timeline, we have only completed a $\textbf{partial}$ hyperparameter sweep so far. Nevertheless, this already improves the TD+CT result on AFHQv2 to $12.87$ (vs. $12.97$ for CT), and the same hyperparameter setting also yields a larger improvement on FFHQ, achieving $15.93$ (vs. $19.45$ for CT). We are continuing the sweep and will include the final tuned result in the camera-ready revision.
>
> **W2(i) On the bootstrap target / target network.**
>
> Yes, this is exactly the case. The target network here is meant to play the same role as the EMA target used in CT. The reason we introduce it is to make the TD construction follow the standard RL intuition more closely: in temporal-difference learning, the target at the current step is formed by a one-step reward plus an estimate of the value at a previous step, rather than by a fully observed terminal target alone. In our diffusion setting, this “bootstrap target” corresponds to the one-step diffusion drift term with the target-network estimate of the earlier-step value/drift. Conceptually, this is how we connect the RL formulation to our main idea: instead of enforcing consistency only locally or only through a shared endpoint, we use a TD-style target to encourage the multi-step evolution induced by the denoiser to be consistent across time. We will revise the paper to make this connection more explicit.
>
> **W2(ii)(iii). On Eq. (6) vs Eq. (9), and the notation $G_t$ vs $g_t$.**
>
> Thank you for pointing this out. We agree that the notation here could be better clarified.
>
> Regarding Eq. (6) and Eq. (9), the intended logic is the following. Eq. (6) introduces the generic RL/MRP-style value definition as the expected return from a state at time $t$. In our diffusion-specific MRP, however, the process is specified relative to a given clean sample $x_0$​, and each reward is defined with respect to that same $x_0$. Under this conditioning, $x_0$​ is treated as fixed rather than as an additional random variable, and the return becomes deterministic. Eq. (9) is meant to be exactly this diffusion-specific instantiation of the abstract value/return in Eq. (6). We agree that this transition was not explained clearly enough, and we will revise the text to make the conditioning on $x_0$​ explicit in Eq. (6) and to explain carefully how the generic RL formulation maps to our diffusion objective.
>
> $G_t$ denotes the standard RL-style return, which is typically scalar-valued. In our diffusion instantiation, however, the reward/return becomes vector-valued because it is defined in terms of posterior-mean differences in data space. For this reason, we switch to $\mathbf{g}_t$ in the diffusion derivation. We will revise the notation to make this distinction explicit.
>
> **W2(iv)(v). Minor presentation issues.**
>
> We agree that the Number of Function Evaluations (NFE) should be defined explicitly, and we will fix this. We will also fix the abstract formatting to comply with the ICML instructions.
>
> **Q1&2. On using only the TD loss.**
>
> In principle, the TD objective can be used on its own. In practice, we combine it with the base loss (e.g., EDM or CT) because the base objective provides a strong local denoising anchor, while the TD term adds cross-time consistency. This makes optimization more stable and typically speeds up convergence, especially early in training. A potential downside of using TD alone is that, because it is bootstrapped, training can become less stable and more sensitive to hyperparameters. We will clarify this design choice in the revision.
>
> Overall, we are grateful for these comments. We will revise the paper to improve the explanation of the bootstrap target, make the conditioning and return notation precise around Eq. (6)/(9), define NFE explicitly, and fix the formatting issues. We also appreciate the reviewer’s overall assessment that the method is technically solid and that the presentational issues are not structural.

---

> > ### Author Rebuttal · Reviewer_Ubfh · 2026-04-01
> >
> > I thank the authors for their answers and clarifications. This resolves the presentation weaknesses I had. Moreover, I think the improved results from the parameter sweep improve the significance of the method. For this reason I will adjust my score to accept.

---

### Official Review · Reviewer_UENA · 2026-03-13

**Soundness:** 3
**Presentation:** 3
**Significance:** 2
**Originality:** 3
**Overall Recommendation:** 4
**Confidence:** 4

**Summary:**

The paper proposes to solve the issue of lack of cross-time consistency in standard diffusion models by proposing a TD learning objective that penalizes inconsistency in model’s multi-step integration during denoising. The idea is to formulate diffusion process as markov reward process and denoising as a policy evaluation problem. Results show improvement in generation quality across different generative models.

**Compliance With Llm Reviewing Policy:**

Affirmed.

**Final Justification:**

The authors provided clarification on some of the recent works and justification for how the proposed method can still be applied as a drop-in training objective. While still some major experimental gaps remain due to no evaluation on how drop-in TD would help enhance the performance of most recent methods like MeanFlow, Shortcut Models etc., making the paper bit outdated. However, due to the originality of the work, decent mathematical depth and it's drop-in capability, i'd still like increase my score to weak-accept.

**Key Questions For Authors:**

Why didn’t you perform more evaluations for class conditioning and for larger datasets? I think it’ just CIFAR-10 which is evaluated for conditional generation case.

**Limitations:**

I think the paper fails to include recent similar works which have significantly improved generative models for few-step generation. Also popular approaches based on optimal control like Adjoint Matching are not discussed and are not considered as a baseline. The experiments are also not extensive enough and does not consider popular benchmarks such as Imagenet.

**Strengths And Weaknesses:**

Strengths:
The paper looks at the generative modelling from RL perspective which are among relatively few works in the field. The theoretical contribution seems to be sound.
The formulated objective can be used as a drop-in replacement to improve generation quality through cross-time consistency in different diffusion models.

Weaknesses:
The experimentation is insufficient and fails to evaluate on widely evaluated Imagenet benchmarks.
I think this works is bit outdated given the current innovations in generative models such as MeanFlow and Shortcut models for ODE based cases and Adjoint Matching for SDE that can achieve very low FID in few-steps evaluation. Thus these should be included as a baseline comparison. Also there are several RL and optimal-control strategies for generative models which the paper fails to discuss and compare as a baseline.
The paper is bit harder to understand though and i feel it can be made much simpler and clear.

---

> ### Author Rebuttal · Authors · 2026-03-31
>
> We thank the reviewer for the thoughtful feedback and for recognizing the novelty of viewing generative modeling from an RL perspective, the soundness of the theoretical formulation, and the potential of our method as a drop-in objective. The focus of this work is to introduce a flexible TD-based training objective that enforces cross-time consistency of the denoising dynamics, rather than a new standalone few-step training paradigm. We respond to the main concerns below.
>
> **Q. Experimental scope / class-conditional and larger-scale evaluation.**
>
> We agree that broader validation, especially on larger-scale class-conditional benchmarks, would further strengthen the paper. At the same time, we would like to clarify the intended scope of the current submission. The main claim is that the proposed TD objective is a general training regularizer that improves temporal consistency across denoising steps and can be incorporated into different base training paradigms. To isolate this contribution, we evaluated it on two setups (EDM and CT) and across multiple datasets and inference budgets. The current results already show gains or competitive performance on CIFAR-10 $(32 \times 32)$, AFHQv2 $(64  \times 64)$, and FFHQ $(64  \times  64)$, with the clearest advantages in, but not limited to, the few-step regime that motivates the paper. We also agree that a larger-scale evaluation is important. The current empirical scope is primarily up to $64  \times  64$, and broader validation is an important next step.
>
> **Weakness. Recent few-step methods such as MeanFlow / Shortcut.**
>
> We thank the reviewer for pointing out these recent works. We agree they should be discussed more clearly as related work. Our TD objective can be used as a drop-in training objective: it keeps the base denoiser model parameterization unchanged and regularizes the model by penalizing the drift mismatch. In this sense, the method is designed to be used on top of existing diffusion training pipelines and enhances their performance rather than replacing them with a new standalone training framework. For this reason, we have already evaluated the proposed TD consistency under two base paradigms, EDM and CT, across multiple datasets and sampling budgets.
>
> MeanFlow and Shortcut, on the other hand, apply a different model parametrization with two timesteps rather than single step prediction, and they are standalone methods. Since the consistency enforced by our method is different from that of MeanFlow and Shortcut, we view these directions as complementary rather than competitive to each other.
>
> **Weakness. Adjoint Matching and optimal-control baselines.**
>
> We thank the reviewer for pointing out Adjoint Matching and related control-based work. We will discuss them more clearly in the related work. However, Adjoint Matching studies a completely different problem: reward fine-tuning toward a reward-tilted distribution, whereas our method does not optimize an external reward and instead improves intrinsic cross-time consistency of the denoising dynamics. We therefore view it as important related work, but not directly comparable in our context.
>
> **Weakness. Clarity and presentation.**
>
> We agree with the reviewer that the paper can be made clearer. In the revision, we will improve the presentation by: (i) clarifying earlier that our method is not reward-driven RL fine-tuning, but a temporal-consistency regularizer for denoising dynamics; (ii) adding a more intuitive high-level explanation centered on matching model drift to diffusion drift across two times; and (iii) improving the notation and organization, especially the explanation of time indices and the distinction from endpoint-consistency methods.
>
> Overall, we appreciate these suggestions and hope the reviewer will consider that the submission’s main contribution is a general TD-based objective for enforcing cross-time consistency, already supported by the current results across multiple datasets, inference budgets, and two base diffusion-training paradigms.

---

> > ### Author Rebuttal · Reviewer_UENA · 2026-04-04
> >
> > Thank you for the clarification. Could you also confirm whether the reported results are fully reproducible and whether the code will be open-sourced? I believe this work could benefit a wide range of applications, and making it reproducible and accessible would help many researchers for future works.

---

> > > ### Author Response · Authors · 2026-04-06
> > >
> > > Thank you for the follow-up question. Yes, reproducibility is important to us. We have prepared an anonymous repository containing the current training code: https://anonymous.4open.science/r/td4diff_icml-8745/README.md
> > >
> > > At this stage, the repository already includes the implementation. We are still finalizing the hyperparameter sweeps, and once these are completed, we will add the corresponding scripts/configs and the final hyperparameter settings used for the reported experiments. We will also make sure the final version clearly documents the experimental details needed to reproduce the results. We appreciate the suggestion and agree that making the work accessible and reproducible will make it more useful to the community.

---

### Decision · Program_Chairs · 2026-04-30

**Decision:**

Accept (regular)

**Comment:**

This paper approaches generative modelling from a novel perspective that penalizes the difference of predictions between time steps. Initially there were concerns around experimental design, but those were largely resolved in the rebuttal by the authors. There remains somewhat smaller concerns around strong convergence guarantees of the algorithm, but the empirical results make up for these concerns sufficiently.